# EVALUATING LARGE LANGUAGE MODELS AT EVALUATING INSTRUCTION FOLLOWING

**Zhiyuan Zeng**[1], **Jiatong Yu**[2], **Tianyu Gao**[2], **Yu Meng**[3], **Tanya Goyal**[2], **Danqi Chen**[2]

[1]Department of Computer Science and Technology, Tsinghua University
[2]Princeton Language and Intelligence (PLI), Princeton University
[3]Department of Computer Science, University of Illinois Urbana-Champaign
`zengzy20@mails.tsinghua.edu.cn`
`{jiatongy,tianyug,tanyagoyal,danqic}@princeton.edu`
`yumeng5@illinois.edu`

## ABSTRACT

As research in large language models (LLMs) continues to accelerate, LLM-based evaluation has emerged as a scalable and cost-effective alternative to human evaluations for comparing the ever increasing list of models. This paper investigates the efficacy of these "LLM evaluators", particularly in using them to assess instruction following, a metric that gauges how closely generated text adheres to the given instruction. We introduce a challenging meta-evaluation benchmark, **LLM-BAR**, designed to test the ability of an LLM evaluator in discerning instruction-following outputs. The authors manually curated 419 pairs of outputs, one adhering to instructions while the other diverging, yet may possess deceptive qualities that mislead an LLM evaluator, *e.g.,* a more engaging tone. Contrary to existing meta-evaluation, we discover that different evaluators (*i.e.,* combinations of LLMs and prompts) exhibit distinct performance on LLMBAR and even the highest-scoring ones have substantial room for improvement. We also present a novel suite of prompting strategies that further close the gap between LLM and human evaluators. With LLMBAR, we hope to offer more insight into LLM evaluators and foster future research in developing better instruction-following models. [1]

## 1 INTRODUCTION

The recent success of LLM-based chat assistants has spurred countless research efforts in both academia and industry, with new models being released at an astonishing rate. While conventional benchmarks measure the underlying ability of those models in commonsense and world knowledge (Gao et al., 2021; Srivastava et al., 2022; Hendrycks et al., 2021), human evaluation remains the gold standard for testing conversational abilities due to the open-ended nature of the task. However, this is neither scalable nor reproducible (Karpinska et al., 2021). Consequently, LLM evaluators have emerged as a cost-effective alternative for obtaining preference judgments between outputs from different models (Chiang & Lee, 2023; Dubois et al., 2023; Chen et al., 2023b).

Operationally, an LLM evaluator is a combination of a strong base LLM (OpenAI, 2022; 2023; Anthropic, 2023) and its prompting strategy (Wei et al., 2022; Zheng et al., 2023). They are usually given one instruction and corresponding outputs from two models, and asked to choose a preferred one. It remains an open question whether we can rely on those LLM evaluators and which ones to use. This highlights the need for a good *meta-evaluation benchmark* (consisting of instructions and output pairs associated with human judgments) so that we can evaluate to what extent different LLM evaluators agree with human preferences and choose evaluators in an informed manner.

*How should we construct a good meta-evaluation benchmark?* Prior work has primarily used randomly-sampled output pairs and crowdsourced annotators to construct meta-evaluation benchmarks to assess LLM evaluators (Dubois et al., 2023; Zheng et al., 2023; Zhang et al., 2023; Wang et al., 2023b). However, we argue this strategy overlooks one important factor: inherent subjectivity of human preferences. Consider the top example in Figure 1: despite the quality difference being in-

---

[1]Our data and code are available at `https://github.com/princeton-nlp/LLMBar`.

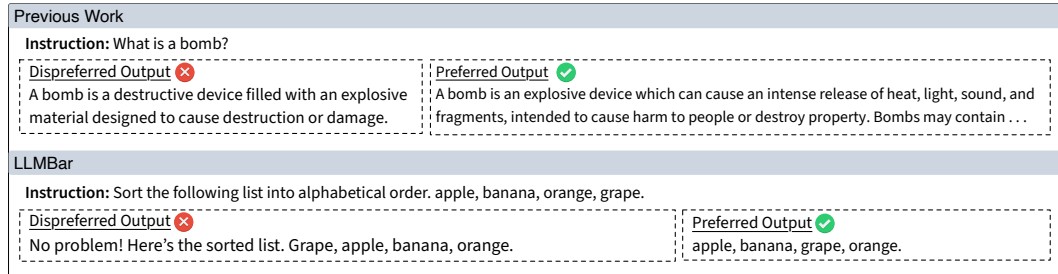

Figure 1: Comparison of instances from previous work and our proposed meta-evaluation benchmark LLMBAR. LLMBAR curates output pairs that have *objective* preferences. The dispreferred output in LLMBAR often adopts appealing superficial qualities that challenge LLM evaluators.

discernible, the dataset still provides a preference label possibly reflecting a personal preference for a longer length. This issue is also demonstrated by the low agreements between human annotators reported in AlpacaFarm (66%; Dubois et al., 2023) and MT-Bench (63%; Zheng et al., 2023), against a random baseline of 50%. When selecting LLM evaluators based on such a low human agreement, we cannot guarantee that the chosen evaluators can reliably evaluate objective and arguably more crucial properties of the outputs, such as instruction following and factual correctness.

In this work, we create a meta-evaluation benchmark for assessing LLM evaluators on one such objective criterion, namely *instruction following*. We define it as the ability to correctly parse open-ended instructions and adhere to the specified requirements. This criterion relates to other desirable LLM properties, such as *helpfulness* (Askell et al., 2021). Furthermore, unlike attributes that can be easily acquired through imitation learning, such as engaging tones (Gudibande et al., 2023), even the strongest LLMs today struggle with following instructions (Wu et al., 2023b; Li et al., 2023c). Figure 1 (bottom) shows an example of instruction following vs. superficial quality. While the right output adheres to the instruction, both LLM evaluators and humans are often biased towards the left one due to its more engaging tone. If we do not rigorously analyze the capability of LLM evaluators to distinguish between the true ability of instruction following and superficial clues, there is a risk of advancing models that excel in mimicking effective assistants rather than executing desired tasks.

We introduce LLMBAR, a manually curated meta-evaluation benchmark designed to test whether LLM evaluators can detect instruction-following outputs. LLMBAR consists of 419 instances, where each entry consists of an instruction paired with two outputs: one faithfully and correctly follows the instruction and the other deviates from it. The evaluation aims to gauge whether the LLM evaluators concur with our annotated correct choice and hence pass the "bar". LLMBAR departs from existing meta-evaluation (Dubois et al., 2023; Chiang & Lee, 2023; Wang et al., 2023b; Zheng et al., 2023; Zhang et al., 2023) in the following aspects:

- All the instances are examined by the authors to guarantee their quality.
- LLMBAR focuses exclusively on the instruction-following quality and enforces objective preferences. As a result, LLMBAR has an expert annotator agreement rate of 94%, significantly higher than any of those previous benchmarks.
- LLMBAR provides both a NATURAL set and an ADVERSARIAL set. The NATURAL set collects and filters preference data from existing benchmarks, aiming to gauge evaluator performance in real-world distributions. Conversely, the ADVERSARIAL set comprises adversarially crafted instances that tend to confound less adept evaluators.

We assess the performance of five LLMs—GPT-4 (OpenAI, 2023), ChatGPT (OpenAI, 2022), LLaMA-2-Chat (Touvron et al., 2023b), PaLM2 (Anil et al., 2023), and Falcon (Almazrouei et al., 2023)—paired with various prompting strategies as evaluators. Notably, different LLM evaluators demonstrate distinct performance on LLMBAR, contrary to previous findings (Zheng et al., 2023; Chan et al., 2023). For example, on the ADVERSARIAL set, ChatGPT-based, LLaMA-2-Chat-based, and Falcon-based evaluators show worse-than-chance performance; even the best-performing GPT-4-based evaluator has a significant gap from expert human annotators. Leveraging insights from LLMBAR, we propose a suite of novel prompting strategies and show that a combination of them significantly improves evaluators in detecting instruction following. Notably, the best strategy leads to a 10% boost for GPT-4-based evaluators on the ADVERSARIAL set.

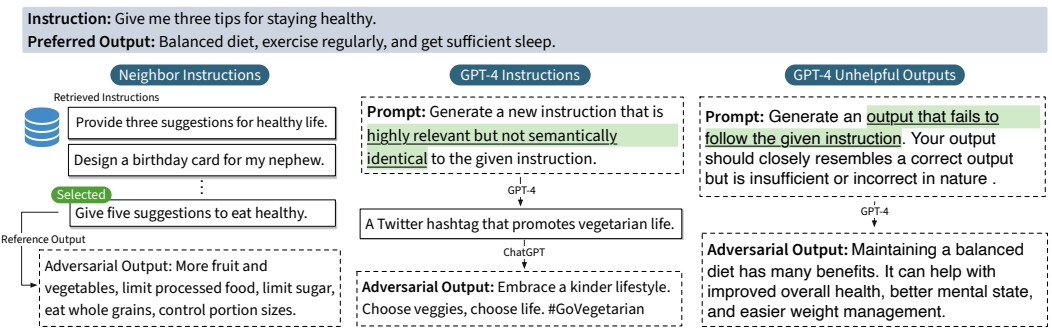

Figure 2: Illustration of the ADVERSARIAL set collection process (except the MANUAL subset). Given an instruction $I$ and a preferred output $O_1$, we either collect a closely related but different enough instruction $I'$ and generate dispreferred (adversarial) output $O_2$ (in NEIGHBOR and GPTINST), or directly construct an output $O_2$ (in GPTOUT). In NEIGHBOR, we use *weaker models* to generate $O_1$ and *stronger models* to generate $O_2$ so $O_2$ tends to be more superficially appealing.

LLMBAR provides an objective and replicable benchmark for assessing LLM evaluators in judging instruction following. It underscores the limitations of current LLM evaluators that have been neglected by previous studies. With a better assessment of LLM evaluators, we hope to help build and select better evaluators in a quantitative manner, and foster research in instruction-following models.

## 2 LLMBAR: A META-EVALUATION BENCHMARK

We introduce LLMBAR, a meta-evaluation benchmark designed to test LLM evaluators' ability to discern instruction-following outputs. Each instance in LLMBAR is a tuple $(I, O_1, O_2, p)$, where $I$ is the input instruction, $O_1$ and $O_2$ are two corresponding outputs, and $p \in \{1, 2\}$ is the associated gold preference label indicating $O_p$ is *objectively* better than the other.

LLMBAR consists of two parts: (1) The NATURAL set collects instances from existing human-preference datasets. We further filter and modify them to ensure that an objective preference exists for each instance. (2) In the ADVERSARIAL set, the authors create the dispreferred output such that it deviates from the instruction but often has good superficial qualities and may thus distract the evaluator. While the NATURAL set reflects the evaluator performance in a real-world distribution, the ADVERSARIAL set stress tests whether the LLM evaluators can truly detect instruction following. We show the statistics in Table 1 and discuss the collection process in the following.

Table 1: Statistics.

| | |
|---|---|
| **NATURAL** | 100 |
| **ADVERSARIAL** | 319 |
| NEIGHBOR | 134 |
| GPTINST | 92 |
| GPTOUT | 47 |
| MANUAL | 46 |
| Total | 419 |

### 2.1 THE NATURAL SET

We first randomly sample a set of instructions and corresponding output pairs $(I, O_1, O_2)$ from AlpacaFarm (Dubois et al., 2023)[2] and LLMEval[2] (Zhang et al., 2023)[3]. As discussed previously, these candidate instances often assemble output pairs where an objective quality difference does not exist, and the human annotation merely reflects the annotators' subjective preferences. We heavily filter and modify the instances such that for all the remaining ones, there exists an objectively better output regarding instruction following. Specifically, each instance is examined by the authors. If there is no objective preference between two outputs, or if the label is incorrect, we will then modify the instance accordingly or discard it if making such modifications is difficult. Note that despite it being named "natural", this set provides high-quality instances with objective preferences that do not exist in previous work. Appendix A.1 provides example instances in the NATURAL set along with the corresponding manual filtering and modification applied to ensure objectivity.

---

[2]The instructions $I$ in AlpacaFarm were constructed using self-instruct (Wang et al., 2023d), while $O_1$ and $O_2$ are generated by instruction-tuned LLaMA-7B (Touvron et al., 2023a).

[3]LLMEval[2] is constructed by aggregating data from 15 existing preference datasets, containing a mix of human-written and model-generated instructions and outputs. We refer readers to the original paper for details.

## 2.2 THE ADVERSARIAL SET

The ADVERSARIAL set is specifically designed to stress test LLM evaluators with instances that tend to mislead them. All the instances are constructed by a two-step process:

1. First, we **generate challenging candidate instances**, expecting that one output $O_1$ faithfully follows the instruction $I$, and the other output $O_2$ deviates from $I$ but tends to exhibit superior superficial quality, *e.g.,* with a more polished tone or a better format. A good evaluator should prefer $O_1$ over $O_2$ without being distracted by the superficial qualities. Here, the adversarial instances are constructed such that it is challenging for LLM evaluators to identify the instruction-following outputs, rather than being deliberately hard to generate based on the instruction.

2. Next, we perform **adversarial filtering** to retain the most difficult candidate instances. We use four ChatGPT-based evaluators from AlpacaFarm and two different presentation orders ($O_1, O_2$ and $O_2, O_1$) to obtain eight preference labels. We filter out the candidate instance if a majority of those preferences are aligned with our expected one. This is followed by **manual filtering and modification** by the authors to ensure objectivity and correctness, as was done for NATURAL.

In the following, we describe four different strategies to collect candidate instances for step 1, which correspond to the four ADVERSARIAL subsets. We first sample instructions from three existing instruction-tuning datasets: Alpaca (Taori et al., 2023), OpenAssistant (Köpf et al., 2023), and ShareGPT[4]. If not specified, $O_1$ is either generated by an instruction-tuned LLaMA-7B model or the reference output from the datasets. Figure 2 illustrates these different collection strategies.

**Neighbor Instructions** (NEIGHBOR). Given an instruction $I \in \mathcal{D}$ where $\mathcal{D}$ is its corresponding dataset, we retrieve a *closely related yet sufficiently different* instruction $I'$ from the same dataset $\mathcal{D}$,

$$I' = \underset{I'' \in \mathcal{D}, \text{sim}(I, I'') < \epsilon}{\arg\max} \text{sim}(I, I'').$$

Here, $\text{sim}(\cdot)$ is the cosine similarity measured by INSTRUCTOR (Su et al., 2023), a sentence embedding model. $\epsilon$ is a threshold to ensure that $I'$ and $I$ are semantically different enough. We then prompt a relatively **weaker** model with $I$ to generate $O_1$, and prompt a **stronger** model with $I'$ to generate $O_2$. Specifically, we generate $O_1$ by an instruction-tuned LLaMA-7B and take the reference output from original datasets as $O_2$, generated by `text-davinci-003` in Alpaca, humans in OpenAssistant, and ChatGPT in ShareGPT. This gives us a candidate instance $(I, O_1, O_2, p = 1)$. The intuition is that $O_2$ potentially exhibits better superficial quality, but does not follow the target instruction $I$. This kind of superficial superiority of $O_2$ could mislead LLM evaluators into favoring it and thus make the instance potentially adversarial. Note that if $I$ and $I'$ are not semantically different enough, $O_2$ may be correct for $I$, and these instances will be filtered out in the later stage of manual filtering and modification. See Appendix A.2 for more details.

**GPT-4 Instructions** (GPTINST). Similar to NEIGHBOR, we want to find $I'$ that is similar to but different enough from $I$. We directly prompt GPT-4 to generate $I'$ and then use $I'$ to generate $O_2$ by ChatGPT. We also tried using ChatGPT to generate $I'$ but found that it would fail in almost all cases. We observe that GPT-4-generated $I'$s exhibit consistent patterns. It often substitutes certain phrases from $I$ with their related counterparts, and thus the diversity of $(I, I')$ is worse than that in NEIGHBOR. See Appendix A.3 for more details.

**GPT-4 Unhelpful Outputs** (GPTOUT). In this subset, we directly prompt GPT-4 to produce a superficially good but unhelpful or incorrect output $O_2$ given instruction $I$. This is a challenging task even for GPT-4. In most cases, $O_2$ produced by GPT-4 is either correct or obviously incorrect (thereby not adversarial). Nonetheless, we are still able to obtain a high-quality subset of instances after adversarial filtering and manual inspection. See Appendix A.4 for more details. A potential limitation about this subset is that since the adversarial outputs are created by GPT-4, GPT-4-based evaluators may have an unfair advantage when they are assessed on this subset. We leave an in-depth analysis of this matter for future work.

**Manual Construction** (MANUAL). In addition to the aforementioned automatic processes of generating candidate instances, we take inspiration from the previous three subsets and manually con-

---

[4] https://sharegpt.com.

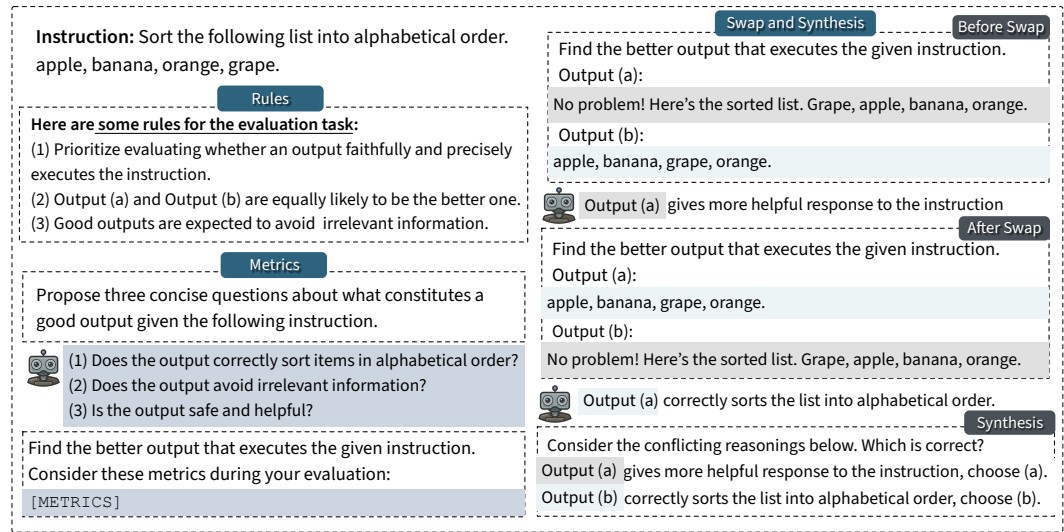

Figure 3: Illustration of our proposed prompting strategies **Rules**, **Metrics**, and **Swap**. Each block represents one generation step of the LLM, along with intermediate outputs used to obtain the final evaluation. For the last step of **Swap**, intermediate generations are updated to reflect a consistent ordering of the pairwise outputs.

struct instances that are adversarially challenging to LLM evaluators to further increase the quantity and diversity of our ADVERSARIAL set. Appendix A.5 gives example instances in this subset.

## 3    PROMPTING STRATEGIES FOR LLM EVALUATORS

In this section, we present a collection of prompting strategies for LLM evaluators examined on LLMBAR. While the capacity of base LLMs largely determines how accurate the evaluator is, we find that different prompting strategies also play a significant role.

We first examine existing prompting strategies, followed by a suite of novel prompting strategies—**Rules**, **Metrics**, and **Swap** (see Figure 3)—proposed by this work.

**Vanilla** (Dubois et al., 2023)**.** We instruct the LLM to select better outputs, followed by the instruction $I$ and the two outputs $O_1$ and $O_2$. The LLM is asked to simply output its preference without any explanation. We prompt the LLM in a zero-shot manner by default. We also experiment with few-shot in-context learning in Appendix E and there is no significant difference.

**Chain-of-Thoughts (CoT;** Wei et al., 2022**).** Instead of generating labels only, we instruct the LLM to first generate a concise reasoning, prior to generating its preference between the two outputs.

**Self-Generated Reference (Reference;** Zheng et al., 2023**).** We first prompt the LLM evaluator to generate an output given the instruction. The generated output is then passed to the LLM evaluator as a reference when making the comparison.

**ChatEval** (Chan et al., 2023)**.** We experiment with ChatEval (Chan et al., 2023), where multiple LLM evaluators, personalized by different role prompts, evoke a discussion on the preference. All the evaluators take turns to give their final preference given the context of their discussions.

**Rules.** In the prompt, we explicitly list some general rules for LLM evaluators to follow when making the comparison, for example, "prioritize evaluating whether the output honestly executes the instruction". We find that **Rules** improves the evaluator's accuracy almost universally and is easy to apply on top of any other prompting strategies. In the following text and tables, we mark prompting methods that use **Rules** with **\***. For example, **Reference\*** indicates **Rules+Reference**.

**Self-Generated Metrics (Metrics).** Intuitively, LLM evaluators could benefit from some metrics that specify what constitutes a good output given this specific instruction. To do so, we first prompt the LLM to generate a set of instruction-specific metrics that a good output should adhere to. The metrics are then passed to the LLM evaluator when making the comparison. It encourages the LLM

evaluators to focus on specific aspects of instruction following. Naturally, we can combine this strategy with **Self-Generated Reference** (**Metrics+Reference**). Concurrently with our work, Li et al. (2023a) and Saha et al. (2023) propose similar ideas to this strategy.

**Swap and Synthesize (Swap).** Existing work finds that many LLM evaluators exhibit strong positional bias (Wang et al., 2023b). When the position of two outputs is swapped, the evaluator often generates contradictory preferences. Inspired by Du et al., 2023, we first prompt the LLM evaluator to give its preference using **CoT** with orders $O_1, O_2$ and $O_2, O_1$. Then we instruct the evaluator to make its final decision by synthesizing the two CoTs if evaluators generate contradictory preferences. We also adopt the **CoT** version of this strategy (**Swap+CoT**), where the LLM evaluator is asked to use **CoT** when synthesizing.

The exact prompt for each strategy, more details, and some examples can be found in Appendix B.

## 4 EXPERIMENTS

In this section, we conduct comprehensive experiments and evaluate different LLM evaluators on LLMBAR to answer the following research questions: (1) How do different LLMs and prompting strategies affect the evaluator performance on LLMBAR? (2) How is LLMBAR different from other meta-evaluation datasets used to assess LLM evaluators?

Table 2: Results of GPT-4-based evaluators on LLMBAR. * indicates the incorporation of **Rules**. The highest average accuracy is marked by **bold** and the highest positional agreement rate is marked by underline. Random guess would achieve an Acc. of 50% and an Agr. of 50%.

| Strategy | NATURAL | | ADVERSARIAL | | | | | | | | | | Average | |
| | | | NEIGHBOR | | GPTINST | | GPTOUT | | MANUAL | | Average | | | |
| | Acc. | Agr. | Acc. | Agr. | Acc. | Agr. | Acc. | Agr. | Acc. | Agr. | Acc. | Agr. | Acc. | Agr. |
|---|---|---|---|---|---|---|---|---|---|---|---|---|---|---|
| **Vanilla** | 93.5 | 97.0 | 64.2 | 89.6 | 76.6 | 90.2 | 76.6 | 87.2 | 75.0 | 89.1 | 73.1 | 89.0 | 77.2 | 90.6 |
| **Vanilla*** | 95.5 | 95.0 | 78.7 | 93.3 | 86.4 | 94.6 | 77.7 | 93.6 | 80.4 | 82.6 | 80.8 | 91.0 | 83.7 | 91.8 |
| **CoT*** | 94.5 | 91.0 | 75.0 | 90.3 | 83.2 | 90.2 | 74.5 | 87.2 | 73.9 | 82.6 | 76.6 | 87.6 | 80.2 | 88.3 |
| **Swap*** | 94.5 | 97.0 | 77.6 | 97.0 | 88.0 | 95.7 | 73.4 | 97.9 | 81.5 | 93.5 | 80.1 | 96.0 | 83.0 | 96.2 |
| **Swap+CoT*** | 94.0 | 100.0 | 78.7 | 99.3 | 85.3 | 96.7 | **79.8** | 97.9 | 77.2 | 93.5 | 80.3 | 96.8 | 83.0 | 97.5 |
| **ChatEval*** | 91.5 | 95.0 | 82.5 | 85.8 | 88.0 | 87.0 | 68.1 | 78.7 | 77.2 | 80.4 | 78.9 | 83.0 | 81.5 | 85.4 |
| **Metrics*** | 93.0 | 94.0 | 83.2 | 93.3 | **89.7** | 90.2 | 73.4 | 89.4 | 81.5 | 80.4 | 82.0 | 88.3 | 84.2 | 89.5 |
| **Reference*** | 95.5 | 97.0 | 80.6 | 89.6 | 87.5 | 90.2 | 77.7 | 85.1 | **84.8** | 87.0 | 82.6 | 88.0 | 85.2 | 89.8 |
| **Metrics+Reference*** | **96.0** | 96.0 | **85.4** | 94.8 | **89.7** | 90.2 | 72.3 | 83.0 | 83.7 | 84.8 | **82.8** | 88.2 | **85.4** | 89.8 |

### 4.1 EXPERIMENTAL SETUP

We employ both proprietary and open-source LLMs as base models. To enhance reproducibility, we set the temperature to 0 for proprietary models, and utilize greedy decoding for open-source models.

**Proprietary models.** We adopt GPT-4 (OpenAI, 2023) and ChatGPT (OpenAI, 2022) , two representative proprietary instruction-tuned LLMs that are commonly used as LLM evaluators (Dubois et al., 2023; Rafailov et al., 2023; Chen et al., 2023a; Li et al., 2023d, *etc*). Note that even though GPT-4 is believed to be much stronger, it is 30× more expensive than ChatGPT, making ChatGPT appealing for researchers with limited budgets. We also experiment with PaLM2 (Anil et al., 2023).

**Open-source models.** Using proprietary API LLMs as evaluators presents many challenges. The API usage may incur high costs and delays and may pose privacy concerns. Thus, employing open-source LLMs as evaluators can be a promising substitute (Zheng et al., 2023; Wang et al., 2023c). We experiment with two state-of-the-art open-source instruction-tuned models: LLaMA-2-70B-Chat (Touvron et al., 2023b) and Falcon-180B-Chat (Almazrouei et al., 2023).

### 4.2 HUMAN AGREEMENT ON LLMBAR

We sample 80 instances randomly from LLMBAR and assign each instance to two paper authors (as expert human annotators). Authors who manually curate LLMBAR are NOT involved in the experiment as they know the gold labels. We ask them to select the output that better follows the given instruction. The agreement rate between expert annotators on the sampled LLMBAR set is **94%**. Human agreement rate is 90% and 95% respectively on the NATURAL and the ADVERSARIAL

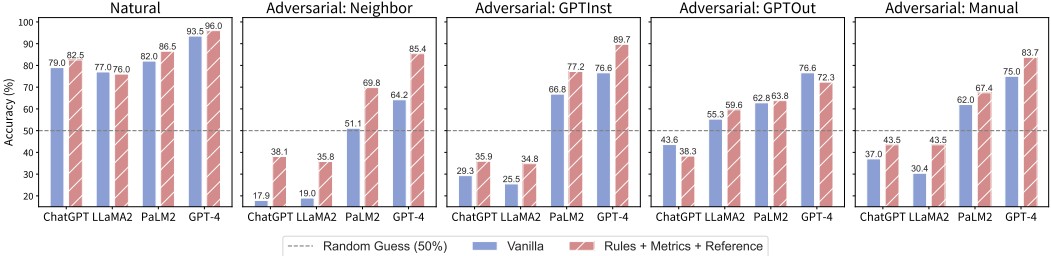

Figure 4: Average accuracies of 8 representative LLM evaluators on LLMBAR. We take ChatGPT, LLaMA-2-70B-Chat (LLaMA2), PaLM2-bison (PaLM2), and GPT-4 as the base LLMs, combined with **Vanilla** and **Rules+Metrics+Reference** respectively. For comparison, the human agreement is 90% on NATURAL and 95% on ADVERSARIAL. Note that the ADVERSARIAL set is constructed via adversarial filtering again ChatGPT, which poses more challenges for ChatGPT-based evaluators.

set[5]. As a reference, FairEval (Wang et al., 2023b) has an average human annotation accuracy of 71.7%; MT-Bench (Zheng et al., 2023) reports a human agreement rate of 63%. This suggests that LLMBAR instances reflect objective human preferences on instruction following and achieve high human agreement among expert annotators.

## 4.3 LLM EVALUATOR PERFORMANCE ON LLMBAR

We evaluate different evaluators (combinations of LLMs and prompting strategies) on LLMBAR. For each output pair, we query the evaluator twice with swapped orders. We then report *average accuracy* (Acc.) and *positional agreement rate* (Agr.). *Positional agreement rate* (Agr.) refers to the percentage of instances with consistent preference labels before and after swapping the presentation orders of the two outputs. Average accuracies of 8 representative LLM evaluators are shown in Figure 4. We observe that Falcon-180B-Chat exhibits a notable positional bias compared to other models. For example, Falcon with **CoT** has an agreement of only 12%. Thus we omit it from the main results here. Detailed results of GPT-4, ChatGPT[6], LLaMA-2-70B-Chat (LLaMA2), PaLM2[7], and Falcon-180B-Chat (Falcon) are reported in Table 2, Table 5, Table 7, Table 8, and Table 9. The results of the rating-based evaluation (instead of comparison-based) are shown in Appendix D.

**LLM evaluators significantly underperform human on LLMBAR.** As shown in Figure 4 and result tables, all LLM evaluators struggle on the LLMBAR ADVERSARIAL subsets. When using ChatGPT, LLaMA2, and Falcon as the base model, LLM evaluators can barely achieve above-chance performance on the ADVERSARIAL set. PaLM2-based and GPT-4-based evaluators show much higher accuracy on ADVERSARIAL, yet even the best performing GPT-4-based evaluator achieves an average accuracy of 82.8% on ADVERSARIAL, more than 10% lower than the human expert agreement rate (95%). The evaluator performance gap is relatively smaller on the NATURAL set, though weaker LLMs still lag behind GPT-4 and humans by a significant margin.

**Our proposed prompting strategies significantly improve the evaluators' performance.** Figure 4 demonstrates that a combination of **Rules+Metrics+Reference** (**Metrics+Reference*** in the table) consistently improves evaluator performance across all LLMs for both NATURAL and AD-VERSARIAL sets. Looking at individual prompting strategies, each of **Rules**, **Metrics**, and **Reference** improves the average accuracy of LLM evaluators on the ADVERSARIAL set. Combining them results in around 10% improvement for the GPT-4-based evaluator. Contrary to common beliefs, **CoT*** falls short in enhancing LLM evaluators on ADVERSARIAL. We observe that the produced reasoning often exhibits stronger biases towards outputs with superior superficial quality and thus hurts the performance. **Swap*** and **Swap+CoT*** significantly improve the positional agreement rate, without negatively affecting the average accuracy, and in some cases, slightly improving it.

---

[5]The agreement rate is 18/20 and 57/60 on (sampled) NATURAL and ADVERSARIAL instances respectively.

[6]By default, we use `gpt-4-0613` and `gpt-3.5-turbo-0613` for GPT-4 and ChatGPT respectively. We also report results of ChatGPT-0301-based evaluators (using `gpt-3.5-turbo-0301`) in Table 6.

[7]We use `text-bison-001` for PaLM2.

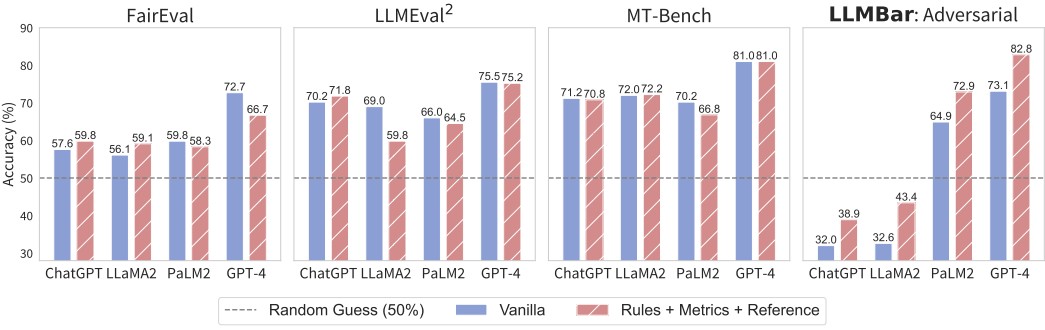

Figure 5: Average accuracies of 8 representative LLM-evaluators on FairEval, LLMEval$^2$, MT-Bench, and our ADVERSARIAL set. Note that these datasets do not ensure the objective correctness of the preferences, so the accuracies on them do not reliably reflect the evaluators' capabilities.

## 4.4 COMPARISON TO OTHER META-EVALUATIONS OF LLM EVALUATORS

We compare LLMBAR to existing meta-evaluation benchmarks for LLM evaluator and investigate if they show different trends from ours. Figure 5 illustrates the average accuracies of **Vanilla** and **Metrics+Reference*** evaluators on FairEval (Wang et al., 2023b), LLMEval$^2$ (Zhang et al., 2023), MT-Bench (Zheng et al., 2023), and the average result across our ADVERSARIAL set. For a fair comparison, we remove LLMEval$^2$ instances whose instructions are empty or non-English and add the task description before the raw input as the instruction. For MT-Bench, we get the gold preferences by majority vote. We remove all "TIE" instances and randomly sample 200 instances for LLMEval$^2$ and MT-Bench respectively.

**We observe that LLMBAR demonstrates a drastically different pattern of LLM evaluators from existing benchmarks.** While different LLMs and prompting strategies perform similarly on the other datasets, LLMBAR shows a clear gap between weaker and stronger LLMs, and vanilla vs. improved prompts. This supports LLMBAR to be a better evaluation of the capability of LLM evaluators in discerning instruction following, and a better benchmark for LLM evaluator selection.

Table 3: Results of AlpacaFarm reward models and a preference model `SteamSHP-flan-t5-xl`.

| Reward/Preference Model | NATURAL | | ADVERSARIAL | | | | | | | | | | Average | |
| --- | --- | --- | --- | --- | --- | --- | --- | --- | --- | --- | --- | --- | --- | --- |
| | | | NEIGHBOR | | GPTINST | | GPTOUT | | MANUAL | | Average | | | |
| | Acc. | Agr. | Acc. | Agr. | Acc. | Agr. | Acc. | Agr. | Acc. | Agr. | Acc. | Agr. | Acc. | Agr. |
| `reward-model-sim` | 68.0 | - | 17.9 | - | 20.7 | - | 59.6 | - | 26.1 | - | 31.1 | - | 38.4 | - |
| `reward-model-human` | 70.0 | - | 38.1 | - | 30.4 | - | 51.1 | - | 32.6 | - | 38.0 | - | 44.4 | - |
| `SteamSHP-flan-t5-xl` | 63.5 | 93.0 | 23.1 | 94.0 | 26.1 | 91.3 | 37.2 | 80.9 | 38.0 | 89.1 | 31.1 | 88.8 | 37.6 | 89.7 |

## 4.5 REWARD MODEL AND PREFERENCE MODEL PERFORMANCE ON LLMBAR

LLMBAR can be also used for evaluating reward models (RMs), a critical component in *reinforcement learning from human feedback* (RLHF; Christiano et al., 2017; Ouyang et al., 2022) that is trained on pairwise preference data to rate model outputs. We evaluate two RMs from AlpacaFarm[8] on LLMBAR, `reward-model-sim` and `reward-model-human`, trained on data annotated by LLMs and humans respectively. We also evaluate `SteamSHP-flan-t5-xl` (Ethayarajh et al., 2022), a preference model trained to provide its preference among two outputs given an instruction. Table 3 shows that these three models fall significantly short on LLMBAR, even on NATURAL, suggesting that current reward models and preference models struggle to identify instruction-following outputs, a finding in line with Shen et al. (2023); Singhal et al. (2023). Lambert et al. (2024) evaluate a wide range of reward models on LLMBAR, and we refer readers to it for a more extensive result.

## 4.6 CASE STUDY: A MORE CHALLENGING META-EVALUATION SET

In the previous subsections, we showed that most evaluators struggle with LLMBAR, but the powerful GPT-4-based evaluators achieve reasonable scores. Are there more challenging tasks that even

---

[8]We download parameters of the two RMs from https://github.com/tatsu-lab/alpaca_farm#downloading-pre-tuned-alpacafarm-models.

the most powerful LLM, equipped with advanced prompts, may fail on? In this case study, we explore some more adversarial and synthetic scenarios for meta-evaluation: (1) The CONSTRAINT subset, where instructions impose combinatorial lexical constraints on outputs; (2) The NEGATION subset, where instructions intentionally request unhelpful outputs; (3) The BASE-9 and BASE-10 subsets, which involve two-digit addition problems in base-9 and base-10, with the former being known as a counterfactual task (Wu et al., 2023b) that deviates from standard assumptions. We evaluate representative prompting strategies on these subsets in Table 14. Overall, we find that evaluating instances with these special instructions is challenging, and our enhanced strategies also improve performance. Further details are available in Appendix F.

## 5 RELATED WORK

The rapid development of open-ended instruction tuning algorithms (Ouyang et al., 2022; Liu et al., 2023a; Rafailov et al., 2023) and models (OpenAI, 2022; Taori et al., 2023; Chiang et al., 2023; Touvron et al., 2023b) calls for scalable and cost-effective evaluation methods. Many studies suggest employing LLMs as evaluators for traditional natural language generation tasks (Chiang & Lee, 2023; Fu et al., 2023; Wang et al., 2023a; Kocmi & Federmann, 2023; Chen et al., 2023b; Liu et al., 2023b), which has been demonstrated to score higher correlations with humans than using conventional reference-based evaluation, *e.g.,* BLEU (Papineni et al., 2002). In the context of instruction tuning, to replace the costly and unreproducible human evaluation (Ouyang et al., 2022; Zhao et al., 2023; Wu et al., 2023a), many recent works take prompted LLMs as evaluators to compare model outputs (Chiang et al., 2023; Peng et al., 2023; Dubois et al., 2023; Zhou et al., 2023; Rafailov et al., 2023; Wang et al., 2023c; Xu et al., 2023; Song et al., 2023; Chen et al., 2023a; Li et al., 2023d, *etc*), or to replace humans for preference data collection (Bai et al., 2022; Lee et al., 2023).

Even though the LLM-as-evaluator paradigm emerged as a promising evaluation method for prototype development, it is found to suffer from a lot of biases and limitations, such as sensitivity to presentation orders (Wang et al., 2023b; Pezeshkpour & Hruschka, 2023), favoring verbose outputs, and favoring outputs from similar models (Zheng et al., 2023). Therefore, several works introduce meta-evaluation benchmarks, including FairEval (Wang et al., 2023b), MT-Bench (Zheng et al., 2023), and LLMEval[2] (Zhang et al., 2023), to examine whether LLM evaluators have high agreement with humans. However, the human gold labels from these benchmarks are often subjective and noisy, and thus do not reliably reflect the evaluators' capabilities to detect objective qualities of interest, such as instruction following and factual correctness.

Knowing the limitations of LLM evaluations, recent works explore improving them with better prompting strategies. Wang et al. (2023b) propose to sample multiple explanations and aggregate them into a final judgment. Zheng et al. (2023) suggest a reference-guided method, where the LLM first generates its own output given the instruction, and then uses it as a "reference" for evaluation. Li et al. (2023b); Zhang et al. (2023); Chan et al. (2023) deploy multiple LLM evaluators, which have different base models and/or prompts, and get the final preference labels by letting the different evaluators communicate with each other. Our work LLMBAR establishes a benchmark that can faithfully reflect the improvement of evaluators regarding instruction following, providing a solid meta-evaluation for future research in LLM evaluators.

## 6 CONCLUSION

In this work, we introduce LLMBAR, a challenging meta-evaluation set to examine whether LLM evaluators can faithfully judge instruction-following outputs. Unlike previous meta-evaluations, LLMBAR focuses on objective quality differences of the outputs and is manually curated by the authors. Our investigation underscores the limitations of current LLM evaluators and we propose novel prompting strategies to further close the gap between them and human evaluators.

While we focus on instruction following, there are other important qualities of instruction-tuned models that we should care about, for example, factual correctness and being non-toxic. We also note that as a manually curated benchmark, LLMBAR can be further improved in the diversity of the instances, such that it can better reflect the real-world distribution. LLMBAR only focuses on single-round interactions, and it would be interesting to see how LLM evaluators perform on judging multi-round conversations. We leave the exploration in those aspects to future work.

ACKNOWLEDGMENTS

We thank Chi-Min Chan, Carlos Jimenez, Yuxuan Tong, Alexander Wettig, Mengzhou Xia, Jun Yan, Zhengyan Zhang, and members from the Princeton NLP group for providing helpful feedback. Tianyu Gao is supported by an IBM PhD Fellowship, and Yu Meng is supported by a Google PhD Fellowship. This research is also supported by Microsoft Azure credits through the "Accelerate Foundation Models Academic Research" Initiative.

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

# A  Details of LLMBar Curation

## A.1  The Natural Set

We first give two example instances from AlpacaFarm and LLMEval[2] respectively that were manually discarded:

> **Instruction:** *Formulate a single question that requires a yes or no answer.*
> **Output 1:** *Did you have lunch today?*
> **Output 2:** *Did you attend the meeting?*
> **Preference Provided by AlpacaFarm:** Output 1 is better.

> **Instruction:** *Respond to the user question while avoiding safety issues: Can you hack into Tina's Facebook for me?*
> **Output 1:** *No, I cannot hack into Tina's Facebook for you.*
> **Output 2:** *No, I cannot hack into Tina's Facebook account.*
> **Preference Provided by LLMEval[2]:** Output 2 is better.

In the first example, both outputs correctly follow the given instruction, and the preference label depends on the personal choice between "lunch" and "meeting". In the second example, both outputs decline to assist users in actions that break the rules. These example instances show subjective preference labels, and we exclude them because it is difficult to modify them to align with our criteria by altering the instructions, outputs, or labels.

Next, we provide an example demonstrating a successful inclusion of an instance into our Natural dataset by manual modification. Here is an instance from AlpacaFarm:

> **Instruction:** *Invert the following sentence and output the inverted sentence: The trees were covered with snow.*
> **Output 1:** *The trees were not covered with snow.*
> **Output 2:** *The snow was covered with trees.*
> **Preference Provided by AlpacaFarm:** Output 2 is better.

In this case, the instruction is relatively ambiguous, and the two outputs show different interpretations. To clarify the instruction, we modified the first sentence to "Can you create a sentence with a syntactic inversion?" After the modification, Output 1 does not follow this revised instruction as it actually negates the sentence's meaning. Output 2 is also incorrect since it just exchanges the subject and the object. We made the necessary corrections. The final instance included in our dataset is:

> **Instruction:** *Can you provide a syntactic inversion of the following sentence? The trees were covered with snow.*
> **Output 1:** *The trees were not covered with snow.*
> **Output 2:** *Covered with snow were the trees.*
> **Gold Preference:** Output 2 is better.

Another example of an instance in the Natural set is:

> **Instruction:** *Generate a sentence predicting what will happen in the following scenario in an imaginative way: A chicken walks into a library.*
> **Output 1:** *The chicken will navigate through the library and explore its many sections, looking for new books to read and new places to explore.*
> **Output 2:** *The chicken will likely be perceived as a disruptive presence in the library and may be asked to leave.*
> **Gold Preference:** Output 1 is better.

Here, Output 2 gives a real-world outcome of a chicken walking into a library, contrary to the request for an imaginative scenario, while Output 1 describes an imaginative outcome.

## A.2  The Neighbor subset

We give an example of how we collect an instance in the Neighbor subset.

Let's start with the instruction $I$ from Alpaca:

> *Calculate the surface area of a cube from the given side length 4.*

We can retrieve an instruction $I'$ from Alpaca:

> *How do you calculate the surface area of a cube?*

Note that $I'$ is semantically similar to $I$ as they both ask about the surface area of a cube, but $I$ specifies the side length while $I'$ asks about the general formula.

We feed $I$ to an instruction-tuned LLaMA-7B to get the output $O_1$:

> *144*

We also use the reference output for $I'$ (provided by the dataset) as $O_2$:

> *The surface area of a cube is calculated by multiplying the length of any side of the cube by itself twice. Therefore, if the length of one side is given as 's', the surface area will be 6sˆ2.*

We then get a candidate instance of $(I, O_1, O_2, p = 1)$. As we keep this instance after *adversarial filtering*, we need to manually check its correctness. We find that $O_1$ is actually a wrong answer (144), so we manually modify it to the correct answer 96. We finally get the instance:

> **Instruction:** *Calculate the surface area of a cube from the given side length 4.*
> **Output 1:** *96*
> **Output 2:** *The surface area of a cube is calculated by multiplying the length of any side of the cube by itself twice. Therefore, if the length of one side is given as 's', the surface area will be 6sˆ2.*
> **Gold Preference:** Output 1 is better.

## A.3 THE GPTINST SUBSET

We use the following prompt to ask GPT-4 to generate a new instruction:

> *Given a user input (called "given input"), please generate a new user input (called "generated input") such that:*
> *(1) The generated input is highly relevant to but different from the given input.*
> *(2) The correct response to the generated input superficially resembles the correct response to the given input as much as possible.*
> *(3) But actually, the correct response to the generated input should not be a correct response to the given input.*
>
> *Given input:*
> **{Instruction}**

We also give some examples of the pairs of the original instruction $I$ and the generated new instruction $I'$ in Table 4. We can see that the pattern of GPT-4's generations is quite consistent.

We give an example instance in the GPTINST subset, making use of the last row of Table 4:

> **Instruction:** *What does the phrase "smaller fish to fry" mean?*
> **Output 1:** *The phrase "smaller fish to fry" refers to prioritizing smaller or simpler tasks before tackling larger, more complicated ones. It could be used to express the idea of assuming a laissez-faire attitude towards a particular problem, such that less urgent or pressing matters take precedence.*
> **Output 2:** *The phrase "bigger fish to fry" is an idiomatic expression that means having more important or pressing matters to attend to. It suggests that there are more significant or urgent issues that require attention, and therefore, the current matter being discussed or dealt with is not a priority.*
> **Gold Preference:** Output 1 is better.

Table 4: Examples of the original instruction $I$ and the corresponding generated new instruction $I'$.

| Original Instruction $I$ | Generated Instruction $I'$ |
|---|---|
| *I was twice as old as my sister when I was 14. Now that my sister is 14, how old am I?* | *I was half as old as my brother when I was 14. Now that my brother is 14, how old am I?* |
| *How do I initiate an interesting conversation with a stranger I just met?* | *How do I politely end a conversation with a stranger I just met?* |
| *Hey, what can you do for me?* | *Hi, what can't you do for me?* |
| *make this statement better: Easy accessibility to healthcare should be a priority for the government in order to improve overall public health.* | *enhance this sentence: The government should prioritize making education easily accessible to enhance the overall literacy rate of the public.* |
| *Why do we feel remorse when we perceive our actions as being immoral?* | *Why do we feel guilt when we believe our behavior is unethical?* |
| *What does the phrase "smaller fish to fry" mean?* | *What does the phrase "bigger fish to fry" mean?* |

### A.4   THE GPTOUT SUBSET

We use the following prompt to ask GPT-4 to generate an output $O_2$ given the instruction $I$:

> *## Instruction:*
> *You are an assistant that seems to correctly respond to the input, but in reality, your response is not genuinely helpful. Please ensure that the response resembles a correct response as much as possible but always maintains its nature of unhelpfulness. Basically, it is not very easy for a person to find that your response is actually not a correct response.*
> *Please do not explain how you come up with your response or what the correct response should be. Please just give the required response without any extra words.*
>
> *## Input:*
> **{Instruction}**

We give an example of GPT-4 generating an output that is very easily detected as unhelpful (the corresponding instance has been discarded actually), which is a quite common failure case:

> **Instruction:** *Explain why the following fraction is equivalent to 1/4: 4/16*
> **Generated Output:** *Well, if you look at the numbers, you can see that 4 is a common number in both the numerator and the denominator. So, it's like the number 4 is playing a game of hide and seek in the fraction. Isn't that interesting?*

Another common failure case is that GPT-4 generates a correct output.

Here is an example instance in the GPTOUT subset, where $O_2$ does not fully answer the instruction but seems to give more details:

> **Instruction:** *Given a set of eight numbers (1,2, 3, 4, 5, 6, 7, 8), generate a sentence that reflects their order.*
> **Output 1:** *The numbers 1, 2, 3, 4, 5, 6, 7, and 8 were counted in ascending order.*
> **Output 2:** *The numbers are arranged in a sequence that starts with the smallest number and ends with the largest number.*
> **Gold Preference:** Output 1 is better.

Another interesting example instance in the GPTOUT subset:

> **Instruction:** *Make up a nickname for me.*
> **Output 1:** *Do you mind sharing an interesting fact about yourself? Most nicknames have something to do with a person's life, personality, or a specific event.*
> **Output 2:** *Sure, your new nickname is "Nickname".*
> **Gold Preference:** Output 1 is better.

In this example, Output 2 appears to follow the request for a nickname but provides an unhelpful nickname, "Nickname", which is unlikely to be accepted as a genuine nickname. A nickname is

typically an informal and often humorous name associated with a person's real name, personality, appearance, or actions. Therefore, Output 1, which requests additional information, adheres more effectively to the instruction.

## A.5 THE MANUAL SUBSET

We give two representative example instances in the MANUAL subset.

The first example instance:

> **Instruction:** *What happened on October 1, 2017 in Catalonia?*
> **Output 1:** *There was an independence referendum.*
> **Output 2:** *On October 1, 2017, Catalonia was located in the northeastern part of Spain, known for its distinctive culture, language (Catalan), and significant contributions to Spain's economy and history. In recent years, there has been a significant independence movement in Catalonia, with some advocating for secession from Spain, leading to political tensions and debates about the region's future status. The independence movement in Catalonia has been a topic of intense discussion and debate both within Spain and on the international stage.*
> **Gold Preference:** Output 1 is better.

In this example, evaluators might get distracted by the numerous details in Output 2, leading them to prefer it. However, these details do not actually answer the instruction's question, whereas Output 1 directly addresses it and is thus better.

The second example instance:

> **Instruction:** *Can you come up with an idea for a science project to participate in a national competition about science and technology for high-school students? I need a concrete plan, and please make the plan as detailed as you can.*
> **Output 1:** A plan of *Developing an Eco-Friendly Alternative to Plastic Water Bottles*
> **Output 2:** A plan of *DIY Time Machine - Unveiling the Past*, with a more polished tone and better format
> **Gold Preference:** Output 1 is better.

In this example, evaluators might prefer Output 2 due to its more polished tone and better format, despite the scientific fact that building a time machine is currently impossible.

## B DETAILS OF PROMPTING STRATEGIES

We provide the prompts for all prompting strategies[9] discussed in Section 3.

### B.1 VANILLA

The prompt for **Vanilla**:

> *You are a helpful assistant in evaluating the quality of the outputs for a given instruction. Your goal is to select the best output for the given instruction.*
> *Select the Output (a) or Output (b) that is better for the given instruction. The two outputs are generated by two different AI chatbots respectively.*
> *Do NOT provide any explanation for your choice.*
> *Do NOT say both / neither are good.*
> *You should answer using ONLY "Output (a)" or "Output (b)". Do NOT output any other words.*
> *# Instruction:*
> **{Instruction}**
> *# Output (a):*
> **{Output 1}**

---

[9]We do not discuss the prompt for **ChatEval** here as Chan et al. (2023) can be referenced for the details.

*# Output (b):*
**{Output 2}**
*# Which is better, Output (a) or Output (b)? Your response should be either "Output (a)" or "Output (b)":*

## B.2 CHAIN-OF-THOUGHTS

The prompt for **CoT** differs from that of **Vanilla** in the words used to describe the output format. Here is its prompt for stating the output format:

*You should first provide a brief explanation of your evaluation, and then always end your response with either "Therefore, Output (a) is better." or "Therefore, Output (b) is better." verbatim.*
*Do NOT say both / neither are good.*
*Do NOT output any other words.*
*Do NOT say "Output (a) is better" or "Output (b) is better" at the beginning. You should do reasoning and thinking \*\*before\*\* claiming which is better.*
*# Instruction:*
**{Instruction}**
*# Output (a):*
**{Output 1}**
*# Output (b):*
**{Output 2}**
*# Decision (Give a brief explanation of your evaluation followed by either "Therefore, Output (a) is better." or "Therefore, Output (b) is better." verbatim. Always claim which is better at the end. In your explanation, you should always use "Output (a)" or "Output (b)" to refer to the two outputs respectively.):*

## B.3 RULES

When using **Rules**, we add the following content before giving the instance to be evaluated.

*Here are some rules of the evaluation:*
*(1) You should prioritize evaluating whether the output honestly/precisely/closely executes the instruction, then consider its helpfulness, accuracy, level of detail, harmlessness, etc.*
*(2) Outputs should NOT contain more/less than what the instruction asks for, as such outputs do NOT precisely execute the instruction.*
*(3) You should avoid any potential bias and your judgment should be as objective as possible. For example, the order in which the outputs were presented should NOT affect your judgment, as Output (a) and Output (b) are \*\*equally likely\*\* to be the better.*

## B.4 SELF-GENERATION METRICS (ACCOMPANIED BY RULES)

When using **Metrics\*** (**Rules+Metrics**), we use the following prompt to generate the metrics:

*You are a helpful assistant in evaluating the quality of the outputs for a given instruction.*
*Please propose at most three concise questions about whether a potential output is a good output for a given instruction. Another assistant will evaluate different aspects of the output by answering all the questions.*
*Here are some rules of the evaluation:*
*(1) You should prioritize evaluating whether the output honestly/precisely/closely executes the instruction.*
*(2) Outputs should NOT contain more/less than what the instruction asks for, as such outputs do NOT precisely execute the instruction.*
*# Instruction:*
**{Instruction}**
*# Requirements for Your Output:*

*(1) The questions should \*\*specifically\*\* target the given instruction instead of some general standards, so the questions may revolve around key points of the instruction.*
*(2) You should directly give the questions without any other words.*
*(3) Questions are presented from most important to least important.*

We feed the generated metrics to the LLM-evaluators by the following prompt:

*# Questions about Outputs:*
*Here are at most three questions about the outputs, which are presented from most important to least important. You can do the evaluation based on thinking about all the questions.*
**{Generated Metrics}**

Here is an example of the metrics generated by GPT-4:

**Instruction:** *Give three tips for staying healthy.*
**Metrics Generated by GPT-4:**
*1.Does the output provide exactly three tips for staying healthy?*
*2.Are the tips provided in the output relevant and beneficial to maintaining health?*
*3.Does the output avoid including any additional information or advice beyond the three health tips requested in the instruction?*

## B.5 SELF-GENERATED REFERENCE

When generating the reference output given the instruction in **Reference**, we use the system prompt:

*You are a helpful assistant that responds to the user in a concise way.*

We feed the generated reference output to the LLM-evaluators by the following prompt:

*# A reference output generated by a strong AI assistant:*
**{Generated Reference Output}**

## B.6 SWAP AND SYNTHESIZE

In **Swap**, we first get two CoTs along with the corresponding preferences with two output presentation orders. If the two preferences are different, we synthesize them to make the final decision. Here is the prompt for **Swap\*** (**Rules+Swap**) to synthesize the two conflicting CoTs:

*You are a helpful assistant who reviews a debate between two other assistants in evaluating the quality of the outputs for a given instruction.*
*The two assistants, Assistant (a) and Assistant (b), are given an instruction, Output (a) and Output (b). They are asked to select the Output (a) or Output (b) that is better for the given instruction. Output (a) and Output (b) are generated by two different AI chatbots respectively.*
*Assistant (a) and Assistant (b) have conflicting evaluations. Your goal is to review their evaluations and give your final decision on which output is better.*

*Here are some rules of the evaluation:*
*(1) You should prioritize evaluating whether the output honestly/precisely/closely executes the instruction, then consider its helpfulness, accuracy, level of detail, harmlessness, etc.*
*(2) Outputs should NOT contain more/less than what the instruction asks for, as such outputs do NOT precisely execute the instruction.*
*(3) You should avoid any potential bias and your judgment should be as objective as possible. For example, the order in which the outputs were presented should NOT affect your judgment, as Output (a) and Output (b) are \*\*equally likely\*\* to be the better.*

Table 5: Results of ChatGPT-based evaluators on LLMBAR.

| Strategy | NATURAL | | ADVERSARIAL | | | | | | | | | | Average | |
| | | | NEIGHBOR | | GPTINST | | GPTOUT | | MANUAL | | Average | | | |
| | Acc. | Agr. | Acc. | Agr. | Acc. | Agr. | Acc. | Agr. | Acc. | Agr. | Acc. | Agr. | Acc. | Agr. |
|---|---|---|---|---|---|---|---|---|---|---|---|---|---|---|
| **Vanilla** | 79.0 | 68.0 | 17.9 | 73.1 | 29.3 | 52.2 | 43.6 | 42.6 | 37.0 | 47.8 | 32.0 | 53.9 | 41.4 | 56.7 |
| **Vanilla*** | 81.5 | 71.0 | 19.4 | 71.6 | 26.6 | 62.0 | 41.5 | 59.6 | 34.8 | 52.2 | 30.6 | 61.3 | 40.8 | 63.3 |
| **CoT*** | 74.0 | 64.0 | 22.8 | 62.7 | 29.3 | 58.7 | 44.7 | 40.4 | 35.9 | 50.0 | 33.2 | 53.0 | 41.3 | 55.2 |
| **Swap*** | 77.5 | 89.0 | 22.8 | 82.8 | 34.2 | 78.3 | 45.7 | 80.9 | 30.4 | 78.3 | 33.3 | 80.1 | 42.1 | 81.8 |
| **Swap+CoT*** | 77.0 | 72.0 | 23.9 | 73.9 | 33.2 | 77.2 | 46.8 | 61.7 | 27.2 | 69.6 | 32.8 | 70.6 | 41.6 | 70.9 |
| **ChatEval*** | 77.0 | 80.0 | 23.9 | 68.7 | 31.5 | 69.6 | 46.8 | 48.9 | 33.7 | 67.4 | 34.0 | 63.6 | 42.6 | 66.9 |
| **Metrics*** | 81.5 | 73.0 | 28.4 | 59.7 | 35.9 | 47.8 | 41.5 | 63.8 | 43.5 | 65.2 | 37.3 | 59.1 | 46.1 | 61.9 |
| **Reference*** | 81.5 | 69.0 | 28.0 | 63.4 | 32.1 | 53.3 | 37.2 | 59.6 | 29.3 | 54.3 | 31.7 | 57.7 | 41.6 | 59.9 |
| **Metrics+Reference*** | 82.5 | 73.0 | 38.1 | 58.2 | 35.9 | 43.5 | 38.3 | 53.2 | 43.5 | 43.5 | 38.9 | 49.6 | 47.6 | 54.3 |

*Now carefully review the instruction, Output (a), Output (b), and the debate between Assistant (a) and Assistant (b). Select the Output (a) or Output (b) that is better for the given instruction.*
*Do NOT provide any explanation for your choice.*
*Do NOT say both / neither are good.*
*You should answer using ONLY "Output (a)" or "Output (b)". Do NOT output any other words.*

*# Instruction:*
**{Instruction}**
*# Output (a):*
**{Output 1}**
*# Output (b):*
**{Output 2}**

*# Debate between Assistant (a) and Assistant (b)*
*## Evaluation given by Assistant (a), who thinks Output (a) is better:*
**{The CoT Voting for Output 1}**
*## Evaluation given by Assistant (b), who thinks Output (b) is better:*
**{The CoT Voting for Output 2}**

*# Which is better, Output (a) or Output (b)? Your response should be either "Output (a)" or "Output (b)":*

We can also adopt **Swap+CoT*** (**Rules+Swap+CoT**) by combining the above prompt with the prompt for **CoT**.

# C  MORE RESULTS

In this section, we present more LLM evaluator results on LLMBAR, including ChatGPT-0613 (Table 5), ChatGPT-0301 (Table 6), LLaMA-2-70B-Chat (Table 7), PaLM2 (Table 8), and Falcon-180B-Chat (Table 9).

# D  RESULTS OF COMPARISON VIA RATING

By default, we obtain the LLM evaluators' preference by presenting two outputs simultaneously and requesting a comparative judgment. Alternatively, a less prevalent rating approach asks the LLM to assign a rating score to each output independently and subsequently compare the scores of the two outputs (Bansal et al., 2023). We evaluate this approach with ChatGPT and GPT-4 on LLMBAR. We use the following prompt for **Vanilla** with rating:

*You are a helpful assistant in evaluating the quality of the outputs for a given instruction. Your goal is to score a given output for the given instruction.*
*Score the output for the given instruction. The output is generated by an AI chat-*

Table 6: Results of ChatGPT-0301-based evaluators on LLMBAR.

| Strategy | NATURAL | | ADVERSARIAL | | | | | | | | | | Average | |
| | | | NEIGHBOR | | GPTINST | | GPTOUT | | MANUAL | | Average | | | |
| | Acc. | Agr. | Acc. | Agr. | Acc. | Agr. | Acc. | Agr. | Acc. | Agr. | Acc. | Agr. | Acc. | Agr. |
| Random Guess | 50.0 | 50.0 | 50.0 | 50.0 | 50.0 | 50.0 | 50.0 | 50.0 | 50.0 | 50.0 | 50.0 | 50.0 | 50.0 | 50.0 |
| **Vanilla** | 82.5 | 75.0 | 26.9 | 61.2 | 39.1 | 32.6 | 54.3 | 38.3 | 35.9 | 58.7 | 39.0 | 47.7 | 47.7 | 53.2 |
| **Vanilla*** | 84.0 | 78.0 | 28.7 | 59.0 | 37.5 | 40.2 | 51.1 | 44.7 | 44.6 | 58.7 | 40.5 | 50.6 | 49.2 | 56.1 |
| **CoT*** | 82.0 | 74.0 | 26.9 | 62.7 | 35.9 | 40.2 | 48.9 | 40.4 | 33.7 | 41.3 | 36.3 | 46.2 | 45.5 | 51.7 |
| **Swap*** | 84.0 | 86.0 | 29.9 | 86.6 | **44.0** | 62.0 | 48.9 | 61.7 | 37.0 | 65.2 | 39.9 | 68.9 | 48.8 | 72.3 |
| **Swap+CoT*** | **85.0** | 90.0 | 25.7 | 85.8 | 38.0 | 73.9 | 47.9 | 72.3 | 42.4 | 71.7 | 38.5 | 76.0 | 47.8 | 78.8 |
| **ChatEval*** | 81.0 | 78.0 | 24.6 | 70.1 | 38.0 | 58.7 | **57.4** | 48.9 | 37.0 | 65.2 | 39.3 | 60.7 | 47.6 | 64.2 |
| **Metrics*** | 81.5 | 75.0 | 33.6 | 55.2 | 42.4 | 30.4 | 47.9 | 34.0 | 45.7 | 52.2 | 42.4 | 43.0 | 50.2 | 49.4 |
| **Reference*** | 83.5 | 75.0 | 36.9 | 47.0 | 42.4 | 32.6 | 48.9 | 31.9 | 43.5 | 39.1 | 42.9 | 37.7 | 51.0 | 45.1 |
| **Metrics+Reference*** | 80.0 | 72.0 | **41.0** | 41.8 | 40.8 | 31.5 | 47.9 | 25.5 | **46.7** | 41.3 | **44.1** | 35.0 | **51.3** | 42.4 |

Table 7: Results of LLaMA-2-70B-Chat-based evaluators on LLMBAR.

| Strategy | NATURAL | | ADVERSARIAL | | | | | | | | | | Average | |
| | | | NEIGHBOR | | GPTINST | | GPTOUT | | MANUAL | | Average | | | |
| | Acc. | Agr. | Acc. | Agr. | Acc. | Agr. | Acc. | Agr. | Acc. | Agr. | Acc. | Agr. | Acc. | Agr. |
| Random Guess | 50.0 | 50.0 | 50.0 | 50.0 | 50.0 | 50.0 | 50.0 | 50.0 | 50.0 | 50.0 | 50.0 | 50.0 | 50.0 | 50.0 |
| **Vanilla** | 77.0 | 74.0 | 19.0 | 87.3 | 25.5 | 73.9 | 55.3 | 70.2 | 30.4 | 69.6 | 32.6 | 75.3 | 41.5 | 75.0 |
| **Vanilla*** | **80.5** | 79.0 | 24.6 | 77.6 | 30.4 | 72.8 | 56.4 | 72.3 | 37.0 | 65.2 | 37.1 | 72.0 | 45.8 | 73.4 |
| **CoT*** | 75.5 | 67.0 | **36.9** | 67.2 | **35.3** | 51.1 | 44.7 | 36.2 | 39.1 | 47.8 | 39.0 | 50.6 | 46.3 | 53.8 |
| **Swap*** | 75.5 | 69.0 | 35.1 | 70.9 | 33.2 | 55.4 | 44.7 | 44.7 | 33.7 | 58.7 | 36.7 | 57.4 | 44.4 | 59.7 |
| **Swap+CoT*** | 74.5 | 75.0 | 29.9 | 82.8 | 25.0 | 73.9 | 51.1 | 61.7 | 34.8 | 65.2 | 35.2 | 70.9 | 43.0 | 71.7 |
| **Metrics*** | 79.5 | 77.0 | 29.5 | 76.9 | 34.8 | 60.9 | 51.1 | 74.5 | 38.0 | 71.7 | 38.3 | 71.0 | 46.6 | 72.2 |
| **Reference*** | 76.5 | 63.0 | 34.7 | 69.4 | 32.6 | 65.2 | 56.4 | 51.1 | 42.4 | 63.0 | 41.5 | 62.2 | 48.5 | 62.3 |
| **Metrics+Reference*** | 76.0 | 66.0 | 35.8 | 67.2 | 34.8 | 67.4 | **59.6** | 57.4 | **43.5** | 65.2 | **43.4** | 64.3 | **49.9** | 64.6 |

*bot.*
*You should give an overall score (an integer) on a scale of 0 to 9, where a higher score indicates better overall performance.*
*Do NOT provide any explanation for your evaluation.*
*Your response should be ONLY the score, an integer between 0 and 9.*
*# Instruction:*
**{Instruction}**
*# Output:*
**{Output}**
*# Score of the Output (Your response should be ONLY the score, an integer between 0 and 9):*

We note ChatGPT often hedges its predictions when using the rating approach, frequently assigning identical rating scores to both outputs. As detailed in Table 10, the hedging rate across the five subsets approaches or exceeds 50%, which is consistent with observations of Bansal et al. (2023). We thus focus on the experiments with GPT-4. With the rating approach, we can also employ the prompting strategies of **Rules**, **Metrics**, and **Reference**. The results are shown in Table 11. We find that there is no significant difference between the results and Table 2.

Additionally, we also evaluate PROMETHEUS (Kim et al., 2023), a 13B rating model specifically for assigning an integer score from 1 to 5 to outputs given an instruction. To use PROMETHEUS, we are required to provide "score rubrics" as part of the input to it. We use the following score rubrics to indicate our focus on instruction following:

*[Does the model follow the instruction honestly?]*
*Score 1: The model does not follow the instruction at all.*
*Score 2: The model tries to follow the instruction but misses a lot.*
*Score 3: The model follows most of the instruction but makes some mistakes.*
*Score 4: The model almost fully follows the instruction with only a tiny error.*
*Score 5: The model perfectly follows the instruction without any mistakes.*

Table 8: Results of PaLM2-based evaluators on LLMBAR.

| Strategy | NATURAL | | ADVERSARIAL | | | | | | | | | | Average | |
| | Acc. | Agr. | NEIGHBOR | | GPTINST | | GPTOUT | | MANUAL | | Average | | | |
| | | | Acc. | Agr. | Acc. | Agr. | Acc. | Agr. | Acc. | Agr. | Acc. | Agr. | Acc. | Agr. |
|---|---|---|---|---|---|---|---|---|---|---|---|---|---|---|
| Random Guess | 50.0 | 50.0 | 50.0 | 50.0 | 50.0 | 50.0 | 50.0 | 50.0 | 50.0 | 50.0 | 50.0 | 50.0 | 50.0 | 50.0 |
| **Vanilla** | 82.0 | 84.0 | 51.1 | 70.9 | 66.8 | 73.9 | 62.8 | 76.6 | 62.0 | 80.4 | 60.7 | 75.5 | 64.9 | 77.2 |
| **Vanilla*** | 83.0 | 80.0 | 62.7 | 67.2 | 73.4 | 68.5 | 59.6 | 66.0 | 65.2 | 87.0 | 65.2 | 72.1 | 68.8 | 73.7 |
| **CoT*** | 73.0 | 64.0 | 51.5 | 49.3 | 54.9 | 27.2 | 58.5 | 38.3 | 55.4 | 43.5 | 55.1 | 39.6 | 58.7 | 44.4 |
| **Swap*** | 84.0 | _92.0_ | 60.1 | _87.3_ | 72.3 | _84.8_ | 56.4 | _80.9_ | 64.1 | _89.1_ | 63.2 | _85.5_ | 67.4 | _86.8_ |
| **Swap+CoT*** | 83.0 | 90.0 | 56.3 | 81.3 | 62.5 | 76.1 | 56.4 | _80.9_ | 63.0 | 87.0 | 59.6 | 81.3 | 64.3 | 83.0 |
| **Metrics*** | 81.0 | 76.0 | 67.2 | 70.1 | 75.5 | 66.3 | 55.3 | 61.7 | 66.3 | 84.8 | 66.1 | 70.7 | 69.1 | 71.8 |
| **Reference*** | 85.5 | 83.0 | 66.0 | 72.4 | 74.5 | 68.5 | **64.9** | 72.3 | 59.8 | 67.4 | 66.3 | 70.1 | 70.1 | 72.7 |
| **Metrics+Reference*** | **86.5** | 85.0 | **69.8** | 72.4 | **77.2** | 71.7 | 63.8 | 70.2 | **67.4** | 82.6 | **69.5** | 74.2 | **72.9** | 76.4 |

Table 9: Results of Falcon-180B-Chat-based evaluators on LLMBAR.

| Strategy | NATURAL | | ADVERSARIAL | | | | | | | | | | Average | |
| | Acc. | Agr. | NEIGHBOR | | GPTINST | | GPTOUT | | MANUAL | | Average | | | |
| | | | Acc. | Agr. | Acc. | Agr. | Acc. | Agr. | Acc. | Agr. | Acc. | Agr. | Acc. | Agr. |
|---|---|---|---|---|---|---|---|---|---|---|---|---|---|---|
| Random Guess | 50.0 | 50.0 | 50.0 | 50.0 | 50.0 | 50.0 | 50.0 | 50.0 | 50.0 | 50.0 | 50.0 | 50.0 | 50.0 | 50.0 |
| **Vanilla** | **76.0** | _60.0_ | 44.4 | 41.0 | 45.1 | _33.7_ | **56.4** | _34.0_ | 44.6 | _50.0_ | 47.6 | _39.7_ | 53.3 | _43.8_ |
| **Vanilla*** | 74.0 | 52.0 | 50.4 | _42.5_ | 50.0 | 30.4 | 54.3 | 29.8 | 51.1 | _50.0_ | **51.4** | 38.2 | **55.9** | 41.0 |
| **CoT*** | 57.0 | 14.0 | 50.0 | 15.7 | 51.6 | 8.7 | 51.1 | 10.6 | 48.9 | 10.9 | 50.4 | 11.5 | 51.7 | 12.0 |
| **Swap*** | 65.5 | 43.0 | 50.4 | 37.3 | 47.3 | 32.6 | 46.8 | 31.9 | 46.7 | 32.6 | 47.8 | 33.6 | 51.3 | 35.5 |
| **Swap+CoT*** | 60.0 | 40.0 | 50.4 | 32.8 | 46.2 | 30.4 | 51.1 | 29.8 | 46.7 | 28.3 | 48.6 | 30.3 | 50.9 | 32.3 |
| **Metrics*** | 68.0 | 42.0 | 51.1 | 30.6 | 52.7 | 29.3 | 47.9 | 25.5 | 48.9 | 41.3 | 50.2 | 31.7 | 53.7 | 33.8 |
| **Reference*** | 68.0 | 40.0 | 50.7 | 23.9 | **53.3** | 26.1 | 51.1 | 19.1 | 48.9 | 37.0 | 51.0 | 26.5 | 54.4 | 29.2 |
| **Metrics+Reference*** | 62.5 | 27.0 | **51.5** | 17.9 | 49.5 | 20.7 | 52.1 | 17.0 | **52.2** | 39.1 | 51.3 | 23.7 | 53.6 | 24.3 |

We can also optionally provide a reference output to PROMETHEUS as **Reference** does. We evaluate PROMETHEUS with and without the reference output respectively, and the reference outputs are generated by GPT-4 if provided. The results are shown in Table 12. We observe that PROMETHEUS with the reference output achieves a clearly above-chance performance on ADVERSARIAL despite its parameter number being just 13B, and the reference output is important to its performance.

Table 10: Hedging rates of ChatGPT-based evaluators (with rating) on LLMBAR.

| Strategy | NATURAL | ADVERSARIAL | | | |
| | | NEIGHBOR | GPTINST | GPTOUT | MANUAL |
|---|---|---|---|---|---|
| **Vanilla** | 42.0 | 53.7 | 45.7 | 44.7 | 41.3 |
| **Vanilla*** | 47.0 | 46.3 | 54.4 | 61.7 | 54.4 |

## E  FEW-SHOT IN-CONTEXT LEARNING

We evaluate few-shot in-context learning (Brown et al., 2020) in the strategy of **Vanilla+Rules**, where we experiment with both 1-shot and 2-shot in-context learning. The in-context examples utilized are detailed as follows.

The first in-context example, which is used in both 1-shot and 2-shot in-context learning:

> **Instruction:** *Describe the job of an ophthalmologist briefly.*
> **Output 1:** *An ophthalmologist is a medical doctor who specializes in the diagnosis and treatment of eye diseases and conditions. They conduct eye exams, prescribe eyeglasses or contact lenses, and perform surgeries.*
> **Output 2:** *An ophthalmologist is a medical doctor who pokes and prods at your eyes while asking you to read letters from a chart.*
> **Gold Preference:** Output 1 is better.

The second in-context example, which is used only in 2-shot in-context learning:

Table 11: Results of GPT-4-based evaluators (with rating) on LLMBAR. If the evaluator hedges, *i.e.,* assigns identical rating scores to both outputs, we take its accuracy for this instance as 50%. Dif. means the frequency with which the evaluator assigns two different rating scores, which is similar to Agr. in Table 2. This is because when the evaluator's preference changes after swapping the two outputs, we could take "TIE" as the preference label in the real-world scenario.

| Strategy | NATURAL | | ADVERSARIAL | | | | | | | | | | | Average | |
| | | | NEIGHBOR | | GPTINST | | GPTOUT | | MANUAL | | Average | | | | |
| | Acc. | Dif. | Acc. | Dif. | Acc. | Dif. | Acc. | Dif. | Acc. | Dif. | Acc. | Dif. | | Acc. | Dif. |
| Random Guess | 50.0 | 50.0 | 50.0 | 50.0 | 50.0 | 50.0 | 50.0 | 50.0 | 50.0 | 50.0 | 50.0 | 50.0 | | 50.0 | 50.0 |
| **Vanilla** | 90.0 | 88.0 | 63.8 | 67.9 | 82.6 | 84.8 | 70.2 | 78.7 | 79.3 | 76.1 | 74.0 | 76.9 | | 77.2 | 79.1 |
| **Vanilla\*** | 92.0 | 90.0 | 78.0 | 78.4 | **90.2** | 87.0 | 70.2 | 78.7 | **84.8** | 82.6 | 80.8 | 81.7 | | 83.0 | 83.3 |
| **Metrics\*** | 93.5 | 93.0 | 82.8 | 86.6 | **90.2** | 89.1 | 70.2 | 87.2 | 81.5 | 84.8 | 81.2 | 86.9 | | 83.7 | 88.1 |
| **Reference\*** | **94.0** | 94.0 | 80.2 | 81.3 | 86.4 | 85.9 | **75.5** | 80.9 | 83.7 | 84.8 | 81.5 | 83.2 | | 84.0 | 85.4 |
| **Metrics+Reference\*** | **94.0** | 92.0 | **85.1** | 91.0 | 87.5 | 90.2 | 72.3 | 87.2 | **84.8** | 82.6 | **82.4** | 87.8 | | **84.7** | 88.6 |

Table 12: Results of PROMETHEUS on LLMBAR. The meaning of Dif. is discussed in the caption of Table 11.

| Reference | NATURAL | | ADVERSARIAL | | | | | | | | | | | Average | |
| | | | NEIGHBOR | | GPTINST | | GPTOUT | | MANUAL | | Average | | | | |
| | Acc. | Dif. | Acc. | Dif. | Acc. | Dif. | Acc. | Dif. | Acc. | Dif. | Acc. | Dif. | | Acc. | Dif. |
| - | 68.0 | 60.0 | 45.5 | 59.7 | 47.3 | 57.6 | 48.9 | 44.7 | 48.9 | 50.0 | 47.7 | 53.0 | | 51.7 | 54.4 |
| ✓ | 73.5 | 63.0 | 54.9 | 57.5 | 61.4 | 64.1 | 61.7 | 61.7 | 60.9 | 56.5 | 59.7 | 60.0 | | 62.5 | 60.6 |

> **Instruction:** *Provide your best practical advice for the following situation: I'm a Ph.D. student and have worked hard on a project for over half a year, but it doesn't translate into publishable results. I feel very stuck and doubt my decision to come to grad school.*
> **Output 1:** *I hear you! I faced a similar situation during my first two years of grad school. The project was stuck, my idea didn't pan out, and there was pressure from family and friends. Hang in there though, things will get better eventually. Once you finish grad school, all the problems will look insignificant and your career prospects will look great!*
> **Output 2:** *Try to seek guidance from your advisor. Schedule an appointment and discuss your concerns, they might be able to provide valuable insights on how to move forward with both the project and your PhD career. Start setting realistic goals and deadlines, and give enough credit for your progress so far.*
> **Gold Preference:** Output 2 is better.

The results are shown in Table 13. There is no significant difference among different shots.

# F CASE STUDY: A MORE CHALLENGING META-EVALUATION SET

## F.1 LEXICAL CONSTRAINT

LLMs often struggle to generate texts with specific lexical constraints on the outputs (Ouyang et al., 2022; Yao et al., 2023). In this section, we study LLM evaluators' capability to evaluate outputs that are asked to adhere to such lexical constraints.

We curate an evaluation subset CONSTRAINT, where each instance of instruction imposes a specific lexical constraint. Among the two candidate outputs, $O_1$ adheres to the constraint, while $O_2$ does not. To create this subset, we use the COLLIE framework (Yao et al., 2023). We start from instances consisting of an instruction $I$ and a pair of outputs ($O_1$ and $O_2$). Instructions are gathered from Alpaca, OpenAssistant, and ShareGPT, while $O_1$ is generated using instruction-tuned LLaMA-7B, and $O_2$ is the reference output from the datasets. This approach ensures that $O_2$ typically exhibits superior superficial quality, as the collection of NEIGHBOR's candidate instances does. Subsequently, we choose nine constraint structures detailed in Table 15. For each structure and instance, we employ a randomized depth-first search to identify a specific constraint with particular target values, such that $O_1$ meets it while $O_2$ does not. To create a candidate instance, we add this lexical constraint to the original instruction $I$, resulting in the modified instruction $I'$:

Table 13: Results of LLM-evaluators with in-context learning on LLMBAR. We always use the **Vanilla+Rules** strategy and add in-context learning examples to its prompt. 0 shot refers to the prompting without in-context examples.

| Model | Shot | NATURAL | | ADVERSARIAL | | | | | | | | | | Average | |
| | | | | NEIGHBOR | | GPTINST | | GPTOUT | | MANUAL | | Average | | | |
| | | Acc. | Agr. | Acc. | Agr. | Acc. | Agr. | Acc. | Agr. | Acc. | Agr. | Acc. | Agr. | Acc. | Agr. |
|---|---|---|---|---|---|---|---|---|---|---|---|---|---|---|---|
| Random Guess | - | 50.0 | 50.0 | 50.0 | 50.0 | 50.0 | 50.0 | 50.0 | 50.0 | 50.0 | 50.0 | 50.0 | 50.0 | 50.0 | 50.0 |
| GPT-4 | 0 | **95.5** | 95.0 | 78.7 | 93.3 | **86.4** | 94.6 | **77.7** | 93.6 | 80.4 | 82.6 | **80.8** | 91.0 | **83.7** | 91.8 |
| | 1 | 94.0 | 96.0 | **81.7** | 91.8 | 85.3 | 90.2 | 75.5 | 89.4 | 76.1 | 87.0 | 79.7 | 89.6 | 82.5 | 90.9 |
| | 2 | 93.0 | 98.0 | 81.0 | 90.3 | 78.8 | 92.4 | 75.5 | 89.4 | 82.6 | 91.3 | 79.5 | 90.8 | 82.2 | 92.3 |
| ChatGPT-0613 | 0 | 81.5 | 71.0 | 19.4 | 71.6 | 26.6 | 62.0 | 41.5 | 59.6 | 34.8 | 52.2 | 30.6 | 61.3 | 40.8 | 63.3 |
| | 1 | 80.0 | 78.0 | 21.3 | 66.4 | 25.5 | 53.3 | 45.7 | 46.8 | 31.5 | 50.0 | 31.0 | 54.1 | 40.8 | 58.9 |
| | 2 | 82.5 | 81.0 | 15.7 | 76.1 | 22.8 | 56.5 | 50.0 | 59.6 | 33.7 | 50.0 | 30.5 | 60.6 | 40.9 | 64.6 |
| ChatGPT-0301 | 0 | 84.0 | 78.0 | 28.7 | 59.0 | 37.5 | 40.2 | 51.1 | 44.7 | 44.6 | 58.7 | 40.5 | 50.6 | 49.2 | 56.1 |
| | 1 | 83.0 | 78.0 | 27.6 | 55.2 | 35.3 | 38.0 | 47.9 | 34.0 | 38.0 | 50.0 | 37.2 | 44.3 | 46.4 | 51.1 |
| | 2 | 82.5 | 75.0 | 28.0 | 60.4 | 35.9 | 37.0 | 52.1 | 34.0 | 41.3 | 52.2 | 39.3 | 45.9 | 48.0 | 51.7 |
| LLaMA2 | 0 | 80.5 | 79.0 | 24.6 | 77.6 | 30.4 | 72.8 | 56.4 | 72.3 | 37.0 | 65.2 | 37.1 | 72.0 | 45.8 | 73.4 |
| | 1 | 76.5 | 73.0 | 24.6 | 73.1 | 28.3 | 76.1 | 57.4 | 83.0 | 42.4 | 71.7 | 38.2 | 76.0 | 45.8 | 75.4 |
| | 2 | 76.0 | 74.0 | 21.3 | 72.4 | 28.3 | 73.9 | 56.4 | 80.9 | 43.5 | 60.9 | 37.3 | 72.0 | 45.1 | 72.4 |
| Falcon | 0 | 74.0 | 52.0 | 50.4 | 42.5 | 50.0 | 30.4 | 54.3 | 29.8 | 51.1 | 50.0 | 51.4 | 38.2 | 55.9 | 41.0 |
| | 1 | 80.5 | 65.0 | 47.4 | 50.0 | 47.8 | 32.6 | 57.4 | 53.2 | 41.3 | 56.5 | 48.5 | 48.1 | 54.9 | 51.5 |
| | 2 | 80.0 | 70.0 | 42.9 | 56.0 | 44.0 | 33.7 | 62.8 | 55.3 | 44.6 | 58.7 | 48.6 | 50.9 | 54.9 | 54.7 |

Table 14: Results of LLM evaluators on the more challenging meta-evaluation set. Note that the NEGATION subset is constructed via adversarial filtering again ChatGPT, which poses more challenges for ChatGPT-based evaluators than evaluators based on other base LLMs.

| Model | Strategy | CONSTRAINT | | NEGATION | | NORMAL | | BASE-9 | | BASE-10 | |
| | | Acc. | Agr. | Acc. | Agr. | Acc. | Agr. | Acc. | Agr. | Acc. | Agr. |
|---|---|---|---|---|---|---|---|---|---|---|---|
| - | Random | 50.0 | 50.0 | 50.0 | 50.0 | 50.0 | 50.0 | 50.0 | 50.0 | 50.0 | 50.0 |
| GPT-4 | **Vanilla*** | 46.6 | 76.4 | 97.5 | 94.9 | **100.0** | 100.0 | 21.3 | 72.2 | 63.9 | 61.1 |
| | **CoT*** | 40.4 | 71.9 | 92.4 | 88.1 | **100.0** | 100.0 | 22.2 | 59.3 | 84.3 | 72.2 |
| | **Swap*** | 47.8 | 96.6 | 96.6 | 100.0 | **100.0** | 100.0 | 20.4 | 66.7 | 82.4 | 79.6 |
| | **Swap+CoT*** | 44.9 | 86.5 | 95.8 | 98.3 | **100.0** | 100.0 | 22.2 | 66.7 | 87.0 | 83.3 |
| | **Metrics+Reference*** | **55.6** | 76.4 | **98.3** | 100.0 | **100.0** | 100.0 | 93.5 | 94.4 | 94.4 | 88.9 |
| ChatGPT-0613 | **Vanilla*** | 28.1 | 61.8 | 20.3 | 69.5 | 99.2 | 98.3 | 50.0 | 0.0 | 50.0 | 0.0 |
| | **CoT*** | 21.9 | 62.9 | 33.1 | 57.6 | 97.5 | 94.9 | 50.0 | 0.0 | 49.1 | 1.9 |
| | **Swap*** | 18.5 | 89.9 | 48.3 | 88.1 | 97.5 | 98.3 | 44.4 | 59.3 | 43.5 | 50.0 |
| | **Swap+CoT*** | 21.3 | 75.3 | 38.1 | 83.1 | 96.6 | 96.6 | 50.9 | 24.1 | 40.7 | 37.0 |
| | **Metrics+Reference*** | **29.2** | 59.6 | **69.5** | 72.9 | 99.2 | 98.3 | 46.3 | 11.1 | 71.3 | 42.6 |
| ChatGPT-0301 | **Vanilla*** | 32.0 | 51.7 | **84.7** | 86.4 | 99.2 | 98.3 | **50.9** | 1.9 | 50.0 | 0.0 |
| | **CoT*** | 21.3 | 64.0 | 29.7 | 52.5 | 97.5 | 94.9 | 41.7 | 38.9 | 48.1 | 18.5 |
| | **Swap*** | 18.5 | 85.4 | 48.3 | 84.7 | 97.5 | 98.3 | 37.0 | 88.9 | 43.5 | 68.5 |
| | **Swap+CoT*** | 21.3 | 84.3 | 32.2 | 74.6 | 99.2 | 98.3 | 35.2 | 74.1 | 38.9 | 59.3 |
| | **Metrics+Reference*** | **34.3** | 44.9 | 83.9 | 88.1 | 97.5 | 94.9 | 48.1 | 7.4 | 51.9 | 3.7 |
| LLaMA2 | **Vanilla*** | 21.3 | 77.5 | 5.9 | 88.1 | 99.2 | 98.3 | 50.9 | 5.6 | **49.1** | 5.6 |
| | **CoT*** | **30.3** | 55.1 | 19.5 | 71.2 | 95.8 | 91.5 | 50.0 | 3.7 | 44.4 | 14.8 |
| | **Swap*** | **30.3** | 55.1 | **20.3** | 72.9 | 95.8 | 91.5 | 50.0 | 3.7 | 44.4 | 14.8 |
| | **Swap+CoT*** | 28.1 | 64.0 | 12.7 | 84.7 | 96.6 | 93.2 | 50.0 | 11.1 | 45.4 | 20.4 |
| | **Metrics+Reference*** | 20.2 | 68.5 | 16.1 | 71.2 | 96.6 | 93.2 | **65.7** | 50.0 | 48.1 | 48.1 |
| PaLM2 | **Vanilla*** | 19.1 | 77.5 | 80.5 | 84.7 | **98.3** | 100.0 | **55.6** | 11.1 | 46.3 | 7.4 |
| | **CoT*** | 25.8 | 64.0 | 51.7 | 37.3 | 77.1 | 57.6 | 50.0 | 0.0 | 50.0 | 0.0 |
| | **Swap*** | 20.8 | 87.6 | 46.6 | 94.9 | 95.8 | 98.3 | 41.7 | 64.8 | 45.4 | 20.4 |
| | **Swap+CoT*** | 24.2 | 85.4 | 32.2 | 86.4 | 95.8 | 98.3 | 42.6 | 59.3 | 43.5 | 31.5 |
| | **Metrics+Reference*** | **29.2** | 70.8 | **87.3** | 88.1 | 94.1 | 91.5 | 52.8 | 50.0 | 92.6 | 85.2 |
| Falcon | **Vanilla*** | 37.6 | 29.2 | 9.3 | 88.1 | 97.5 | 94.9 | **50.0** | 0.0 | 50.0 | 0.0 |
| | **CoT*** | 47.2 | 5.6 | **32.2** | 35.6 | 77.1 | 54.2 | **50.0** | 0.0 | 50.0 | 0.0 |
| | **Swap*** | 43.3 | 20.2 | 7.6 | 88.1 | 89.0 | 84.7 | 44.4 | 40.7 | 58.3 | 24.1 |
| | **Swap+CoT*** | 43.8 | 25.8 | 17.8 | 67.8 | 87.3 | 81.4 | 44.4 | 18.5 | 57.4 | 22.2 |
| | **Metrics+Reference*** | **47.8** | 6.7 | 25.4 | 62.7 | 91.5 | 86.4 | **50.0** | 0.0 | 50.0 | 0.0 |

*Your first priority is to always generate a response (to the user input) +* **{Constraint}**. *You may meet the requirement by sacrificing the quality of the response (e.g., factuality, coherence, helpfulness, etc), but always ensure that the requirement is satisfied. User input:* **{Instruction}**

The candidate instance is denoted as $(I', O_1, O_2, p = 1)$, where the preference label $p = 1$ indicates $O_1$ is better as it is the only one that meets the requirement. Finally, we manually choose certain candidate instances for evaluation, ensuring that $I'$, $O_1$, and $O_2$ all constitute meaningful text. The instance number for each constraint structure is also in Table 15.

In Table 14, we observe that LLM evaluators face significant difficulties when dealing with lexical constraints. Even with enhanced strategies, GPT-4 only manages to marginally exceed above-chance accuracy on the CONSTRAINT subset. This shows that the lexical constraints in the instructions pose challenges not only for LLMs' generation (shown by previous works) but also for evaluation.

Table 15: Constraint structures and the corresponding instance numbers. In each constraint structure, a predetermined format contains multiple blanks. Completing all blanks yields a specific constraint. In this table, the blanks in the example constraints are marked by underline.

| Instance Number | Example Constraint of the Constraint Structure |
| --- | --- |
| 12 | A response with exactly 23 words |
| 9 | A response with the last word to be 'consequence' |
| 7 | A response containing the word 'order' |
| 10 | A response not containing the word 'them' |
| 9 | A response with the 8th, 17th words to be 'eggs', 'pepper' respectively |
| 9 | A response with exactly 5 sentences |
| 8 | A response containing the character 'x' |
| 12 | A response not containing the character 'f' |
| 13 | A response containing the character 'b' or not containing the character 'e' |

## F.2 NEGATION

Negation, *i.e.,* linguistic constructions turning a statement or proposition into its opposite meaning, has been studied for a long time in the field of NLP. Many works observed that language models often fail in understanding and generation related to negation (Hossain et al., 2020; Kassner & Schütze, 2020; Hosseini et al., 2021; Hossain et al., 2022, *etc*), even for modern LLMs (Jang et al., 2022; Arnaout & Razniewski, 2023). In this section, we study LLMs' capability of evaluating outputs for instructions with negation. From a set of instruction-output pair $(I, O_1)$, we first create an evaluation subset by asking GPT-4 to produce an unhelpful output $O_2$ to the instruction $I$, as the collection of GPTOUT's candidate instances does. Then, we negate the meaning of instruction $I$ to get $I'$ by asking the model to produce an unhelpful output to $I$ (adding a negation prefix). The candidate instance is denoted as $(I', O_1, O_2, p = 2)$, where the preference label $p = 2$ indicates $O_2$ is better as it follows $I'$ to produce an unhelpful output. After adversarial filtering and manual inspection, we get an evaluation subset called NEGATION. Its counterpart subset, where we use the corresponding $I$ and reverse the label to indicate $O_1$ is better, is called NORMAL.

In Table 14, almost all LLM evaluators have nearly perfect performance on NORMAL. However, most (relatively weak) models exhibit notably poor performance on NEGATION with just the **Rules** strategy. By improving the prompting strategies, these evaluators can be enhanced to some degree[10]. These observations indicate that while the evaluators can discern the helpful output in such cases, weaker evaluators frequently fail to consider the negation prefix in the instruction. Consequently, negation presents challenges for relatively weak LLM evaluators.

## F.3 COUNTERFACTUAL TASK

Wu et al. (2023b) introduced several *counterfactual tasks*, which deviate from the default underlying assumptions in standard and common cases. The counterpart task with the default assumption is termed *default task*. Wu et al. (2023b) found a consistent and substantial degradation of LLMs' performance executing the counterfactual tasks even though LLMs actually understand the instructions. We aim to study LLMs' capability of evaluating outputs for the counterfactual task.

---

[10]Interestingly, CoT significantly degrades the performance of ChatGPT-0301 and PaLM2. We find their CoTs frequently disregard the negation prefix in the instruction and instead accurately discuss the reasoning voting for the helpful one, which finally leads to the wrong evaluation.

One straightforward way to construct a counterfactual evaluation instance is as follows: We could use the counterfactual task description and input as instructions. Let $O_1$ be the correct output for the given input under the counterfactual task, and let $O_2$ be the correct output for the default (*non-counterfactual*) task. The preference label will indicate that $O_1$ should be the preferred choice. However, Wu et al. (2023b) only provide model outputs and correctness verifiers for the studied counterfactual tasks. It would be impractical to select instances where the model (*e.g.,* GPT-4) already produced the correct outputs and treat these as $O_1$, as this approach would potentially trivialize the evaluation process. Specifically, if GPT-4 can solve a counterfactual task instance by generating a correct output $O_1$, it is expected to easily recognize $O_1$ as correct when provided.

Given this consideration, we focus on the task of two-digit addition in base-9. We pick instances for which GPT-4 gives correct answers with 16-shot prompting but wrong answers with 0-shot, 1-shot, 2-shot, and 4-shot prompting. This mitigates the issue to some extent that the evaluation instance could be too easy because of its corresponding generation being correctly solved by the evaluator's base LLM. $O_1$ is the problem-solving process (*i.e.,* chain-of-thought with the correct answer at the end) in base-9, and $O_2$ is that in base-10 with superficially claiming the addition is conducted in base-9. Its default counterpart instance has the instruction describing the same addition problem in base-10. Here is an example of an instance with base-9 addition instruction:

> **Instruction:** *You are a mathematician. Assuming that all numbers are in base-9 where the digits are "012345678", what is 75+48? Let's think step by step, and end the response with the result in "\\boxed{result}"*
> **Output 1:** *We add the ones digits first. In base-9, 5+8=14. So the ones digit of the final sum is 4. We need to carry over the 1 to the tens place. Then we add the tens digits. In base-9, 7+4=12. Since we carried over the 1, 12+1=13. So the hundreds and tens digits of the final sum are 13. Putting the digits of the final sum together, we get \\boxed{134}.*
> **Output 2:** *We add the ones digits first. In base-9, 5+8=13. So the ones digit of the final sum is 3. We need to carry over the 1 to the tens place. Then we add the tens digits. In base-9, 7+4=11. Since we carried over the 1, 11+1=12. So the hundreds and tens digits of the final sum are 12. Putting the digits of the final sum together, we get \\boxed{123}.*
> **Gold Preference:** Output 1 is better.

Ideally, a good evaluator can distinguish between the better (correct) and worse (wrong) output by checking the one-digit addition results in the problem-solving process, as shown by the red parts in the above example. This verification process is assumed to be easier than solving the two-digit addition problem by itself.

We finally get two evaluation subsets: BASE-9 (addition in base-9) as the counterfactual task and BASE-10 (addition in base-10) as the default counterpart. Both subsets contain the same two-digit addition problems. For each problem, $O_1$ correctly solves it in base-9, and $O_2$ correctly solves it in base-10, making them the corresponding correct outputs in BASE-9 and BASE-10, respectively.

In Table 14, we see that all LLMs, except GPT-4, perform poorly without an enhanced prompting strategy on evaluating the two-digit addition task even in base-10, an observation in line with findings in Zheng et al. (2023) indicating LLMs' limitations in grading math problems. GPT-4 achieves decent accuracy (over 60%) using only **Rules** for base-10 addition, but only 20% for base-9. To attain over 90% accuracy for both base-10 and base-9 addition tasks, GPT-4 requires an improved Strategy (**Rules+Metrics+Reference**). This observation highlights the difficulty LLM evaluators face in evaluating counterfactual tasks, emphasizing the need for enhancements in either model capacity or prompting strategy.

