# OpenReview forum: "Evaluating Large Language Models at Evaluating Instruction Following"
_ICLR.cc/2024/Conference — ICLR 2024 poster_

### Official Review · Reviewer_65gL · 2023-10-16

**Soundness:** 4 excellent
**Presentation:** 3 good
**Contribution:** 3 good
**Rating:** 8
**Confidence:** 4

**Summary:**

This paper proposes the LLMBar, aiming to evaluate if large language models (LLMs) can serve as an evaluator of the LLMs' instruction-following ability. LLMBar consists of two instruction sets, one collected from other benchmarks that are easy for LLM to identify and another generated with different strategies, incorporating sentence similarity, LLMs, etc, which is hard for LLM to identify. To further improve the ability to evaluate the instruction-following ability, the authors also propose a new prompting strategy by introducing a metric set generated by LLM itself to assist the evaluation. The benchmark experiments are performed on human-level and common open-sourced and proprietary LLMs with various prompting strategies, revealing a distinct ability between different LLMs and showing a significant gap between LLMs and human evaluators on the difficult dataset.

**Strengths:**

1. The soundness of this paper is good. Instruction-following ability is an important ability of LLMs that has not been well-studied. This paper fills the gap in this area by introducing a manually constructed instruction dataset supervised by human annotators.
2. The experiment results are from various common LLMs and prompting strategies, which are enough for understanding if a LLM is a good evaluator.
3. This paper is well-written, easy to follow, and will have a wild interest in the LLM community.

**Weaknesses:**

1. The instructions used for dataset construction contain ambiguous words. For example, the word "imaginative" will have different explanations from different people or LLMs in different aspects; such examples should also be deleted from the dataset.
2. The authors use different prompting strategies for evaluation, but the dataset generation only uses plain prompts. It is interesting yet important to discover how different input instructions enhanced by CoT or other prompting strategies (i.e., $(I_{\text{enhanced}}, O_1, O_2, p)$) affect the judgment of different LLMs.
3. The proposed prompting strategy is not as good as the authors claim (there is no significant improvement on GPT4 and LLaMA2, from my point of view, and the output consistency is also not good enough when changing the order of inputs). I think the authors should rephrase the description of the proposed prompting strategies from different perspectives.

**Questions:**

I found a large performance variance on Base-9 and Base-10. I think this is because the generated metrics largely affect the LLMs' performance. Could authors provide the generated metrics by different LLMs?

Also, the performance of the proposed method on GPT-3.5 and GPT-4 does not seem robust on the GPTOut, I'm curious about whether there exists some bias (or preference) of the generated metrics that affects the performance on different evaluation sets.

---

> ### Author Response · Authors · 2023-11-17
> **Response to Reviewer 65gL [1/3]**
>
> Thank you for your constructive feedback! We agree that a strong capability in evaluating instruction-following is crucial for any LLM to serve as a reliable evaluator, and LLMBar bridges this research gap by investigating it. We believe that LLMBar, by offering high-quality and efficient meta-evaluation, will significantly contribute to the development of LLM evaluators. We will address your comments and suggestions in more detail below.
>
> **(1) The instructions used for dataset construction contain ambiguous words.**
>
> Thanks for raising this point! We acknowledge that some usage of words in the Natural set may seem ambiguous. It is intended as we sample from existing datasets to construct the Natural set and try to keep a real-world distribution. However, we made every effort to make sure that in the provided output pairs, one is objectively better than the other without ambiguity. Taking the “chicken in the library” instance in Appendix A.1 as an example, Output 1 is objectively better than Output 2 as Output 2 is not “imaginative” at all. Such objectivity in preferences is also reflected by the 90% human agreement rate of our Natural set.
>
> **(2) The authors use different prompting strategies for evaluation, but the dataset generation only uses plain prompts.**
>
> If you are suggesting using more sophisticated prompting for dataset generation, especially for the GPTInst subset (as it can change the distribution of instances’ instructions), we appreciate your suggestion and will explore this direction in future work to increase the diversity of our dataset.
>
> **(3) It is interesting yet important to discover how different input instructions enhanced by CoT or other prompting strategies affect the judgment of different LLMs**
>
> Reflecting on your proposed direction, we think this setup (of how instructions are like) is both practical and intriguing: the user's instruction includes "metrics" giving what a good or expected output should entail. As you mentioned, understanding how this kind of instructions influences LLM evaluators is important and interesting.
>
> Given the labor-intensive nature of manually creating such instructions, we propose utilizing LLMs to do that. This practice actually translates into the following prompting strategy: we first generate a list of metrics and then incorporate these metrics into the instruction to create an enhanced instruction $I_\text{enhanced}$. This enhanced instruction, along with the pair of outputs, is then fed into an LLM evaluator (we could use the basic **Vanilla*** prompting strategy here as the simplest version). We refer to this strategy as **Enhanced*** and evaluated it on LLMBar using GPT-4 and ChatGPT respectively. The results of the average accuracies are as follows:
>
> GPT-4:
> |      Strategy      | Natural | Neighbor | GPTInst | GPTOut | Manual |
> | :----------------: | :-----: | :------: | :-----: | :----: | :----: |
> |      Vanilla*      |  95.5   |   78.7   |  86.4   |  77.7  |  80.4  |
> |      Metrics*      |  93.0   |   83.2   |  89.7   |  73.4  |  81.5  |
> |   **Enhanced***    |  92.0   |   81.7   |  88.6   |  71.3  |  77.2  |
> | Metrics+Reference* |  96.0   |   85.4   |  89.7   |  72.3  |  83.7  |
>
> ChatGPT:
> |      Strategy      | Natural | Neighbor | GPTInst | GPTOut | Manual |
> | :----------------: | :-----: | :------: | :-----: | :----: | :----: |
> |      Vanilla*      |  81.5   |   19.4   |  26.6   |  41.5  |  34.8  |
> |      Metrics*      |  81.5   |   28.4   |  35.9   |  41.5  |  43.5  |
> |   **Enhanced***    |  81.5   |   33.2   |  29.3   |  42.6  |  43.5  |
> | Metrics+Reference* |  82.5   |   38.1   |  35.9   |  38.3  |  43.5  |
>
> The results indicate that **Enhanced*** significantly outperforms **Vanilla*** when using ChatGPT as the base LLM. This observation leads to two implications: (1) this strategy appears to effectively assist LLM evaluators, and (2) considering two meta-evaluation instances, even when their instructions convey similar intentions, variations in wording can significantly impact the performance of LLM evaluators. We believe these are both interesting directions for future research.
>
> Note: **Enhanced*** is quite similar to **Metrics***. The difference is that the LLM evaluator now regards the generated metrics as a part of the instance's instruction, instead of as a part of the evaluator's prompt (the prompt instructing LLM evaluators to do evaluation). This can be seen as a way to influence the distribution of instances' instructions. Therefore, it serves as both a strategy and an analysis experiment.

---

> > ### Author Response · Authors · 2023-11-17
> > **Response to Reviewer 65gL [2/3]**
> >
> > **(4) The proposed prompting strategy is not as good as the authors claim. The authors should rephrase the description of the proposed prompting strategies from different perspectives**
> >
> > Thanks for your feedback. On LLMBar’s Adversarial set, the combination of our proposed prompting strategies results in around 10% improvement for both GPT-4 and LLaMA-2-chat (Table 2 and Table 7 in our updated version, comparing row “Vanilla” to row “Metrics+Reference*”), which we believe are “significant”. However, we recognize such descriptions can be ambiguous and we have added quantitative numbers in the updated version.
> >
> > Regarding the agreement rate, our proposed **Swap+CoT*** prompting improves GPT-4’s positional agreement rate to 96.8% on Adversarial, which we believe is a very high number compared to 91.0% achieved by **Vanilla***.
> >
> > **(5) A large performance variance on Base-9 and Base-10. Could authors provide the generated metrics by different LLMs?**
> >
> > After investigating the generated metrics and reference outputs by LLMs on Base-9 and Base-10, we believe it is the generated references that significantly improve the evaluators’ performance, if the base LLM has a decent capability of solving the addition problem (so it can generate reference outputs containing correct answers). The following added results of GPT-4 with **Rules+Metrics** and **Rules+Reference** provide evidence:
> >
> > |        Strategy        | Base-9 Acc. | Base-9 Agr. | Base-10 Acc. | Base-10 Agr. |
> > | :--------------------: | :---------: | :---------: | :----------: | :----------: |
> > |      **Vanilla***      |    21.3     |    72.2     |     63.9     |     61.1     |
> > |      **Metrics***      |    25.0     |    61.1     |    62.96     |    62.96     |
> > |     **Reference***     |    95.4     |    94.4     |     99.1     |     98.1     |
> > | **Metrics+Reference*** |    93.5     |    94.4     |     94.4     |     88.9     |
> >
> > We can see **Metrics** does not bring performance improvement but **Reference** does.
> >
> > Diving deeper, we see that when using **Vanilla**, the evaluators often prefer the output that is correct in Base-10. When using **Metrics**, the metrics are often generic such as the following:
> > ```
> > 1. Does the output accurately execute the instruction by providing a step-by-step calculation of x+y in base-9? 2. Does the output strictly adhere to the base-9 number system, using only the digits "012345678"? 3. Does the output conclude with the final result enclosed in "\\boxed{result}" format as instructed?`, where `x` and `y` are the two numbers to be added, respectively.
> > ```
> > Hence generated metrics do not help much. When using **Reference**, the generated reference often provides the correct answer in Base-9, and the LLM evaluators can make the correct judgement by comparing the outputs’ results with the reference.
> >
> > This is also a common case for non-GPT-4 LLMs evaluating Base-10 addition problems (but not Base-9, because other weaker LLMs cannot provide a good reference output by solving Base-9 correctly).

---

> > > ### Author Response · Authors · 2023-11-17
> > > **Response to Reviewer 65gL [3/3]**
> > >
> > > **(6) The performance of the proposed method does not seem robust on the GPTOut. I'm curious about whether there exists some bias (or preference) of the generated metrics.**
> > >
> > > We agree with your insight!
> > >
> > > In GPTOut, the dispreferred output (generated by asking GPT-4 to produce seemingly helpful but actually unhelpful responses) often possesses a good semantic structure that closely adheres to the instruction, but its content does not follow the given instruction. However, the generated metrics and references often bias LLM evaluators towards the semantic structure of the dispreferred output. This is an important failure case of **Metrics** and **Reference**, presenting a potential direction for improvement in future research. For example, considering the instruction `Create a lesson plan for grade 3 science`, the preferred output is:
> > >
> > > ```
> > > First, let's start with the basics of physics, such as force, motion, and energy. Then, we can move on to biology, exploring the different types of plants and animals found in our environment. Finally, we can delve into earth science, examining the properties of rocks, soil, and water.
> > > ```
> > >
> > > and the dispreferred output is:
> > > ```
> > > Sure, here's a lesson plan for grade 3 science: 1. Introduction: Start the class with a fun science joke. 2. Warm-up: Ask students to share their favorite science fact. 3. Main Lesson: Teach about the life cycle of a unicorn. 4. Activity: Have students draw their own unicorn and label the different stages of its life cycle. 5. Wrap-up: End the class with a unicorn-themed song. 6. Homework: Ask students to research more about unicorns and prepare a short presentation for the next class.
> > > ```
> > >
> > > Despite presenting a well-structured lesson plan with multiple logical stages, the content of the dispreferred output is scientifically wrong (e.g., discussing unicorns, which are fictional) and thus not following the instruction. With GPT-4, the generated metrics are
> > > ```
> > > 1. Does the output provide a detailed and structured lesson plan specifically tailored for grade 3 science? 2. Does the lesson plan cover appropriate topics for a grade 3 science curriculum, and are these topics presented in a logical and progressive order? …
> > > ```
> > > where the emphasis on “detail”, “structure”, “logical”, and “progressive” could inadvertently lead the LLM evaluator to favor the dispreferred output. Moreover, the generated reference output also has a structure very similar to that of the dispreferred output. Influenced by both, GPT-4 fails to provide the correct judgment with **Rules+Metrics+Reference** but outputs correctly with the simpler **Rules+Vanilla** strategy.

---

> > > > ### Comment · Reviewer_65gL · 2023-11-21
> > > > **Response to Rebuttal**
> > > >
> > > > Thanks for the author's hard work in providing further details of extra experiments, evaluation process, and prompting case. My questions are well addressed and I will keep my score unchanged on this good paper.

---

### Official Review · Reviewer_WsKe · 2023-10-31

**Soundness:** 2 fair
**Presentation:** 3 good
**Contribution:** 3 good
**Rating:** 6
**Confidence:** 3

**Summary:**

This paper proposes a challenge meta-evaluator benchmark, LLMBar, used to assess the quality of the LLM-evaluator (LLM + prompt strategies) for instruction following. In addition, the paper provides empirical experiment result on various combination of the LLM model (open-sourced and non-open) and prompt strategies and shows improvement for a novel suite of prompt strategies

**Strengths:**

Strength:
* The paper is overall well-written, easy to follow
* The paper addresses an important current problem of scalable evaluation of the LLM-evaluator’s quality

**Weaknesses:**

Weakness
* There is some confusion in how the evaluation set was generated, for example, in Figure 2. “We often use weaker models to generate O1 and stronger model to generate O2 …”  it is unclear what is weak and strong models are referring. I tried to search that definition in the text on page 4 “Neighbor Instruction”, but still couldn’t find any.
* Similarly, point 2 on page 4, not sure what “manual filtering and modification” entail
* The human evaluators are co-authors of the papers, which can raise issues of objectivity compared with other crowd-sourced human evaluators. What measures are in place to prevent the bias?
* I am wondering whether it is a fair comparison to other benchmarks if the instructions are qualitatively different. For example, it is relatively easy to achieve high inter-rater reliability if the response is factual information rather than subjective
* The experiment result part spent the majority of the session on the performance of the LLM evaluator on the newly proposed benchmark, while only a small section on comparing with other benchmarks. Since the benchmarks are qualitatively different (objective vs subjective), I am not sure whether the result is meaningful. It would be ideal to have another benchmark with comparable features (e.g., objectiveness and similar inter-rater agreement)

**Questions:**

Minor issues to be fixed, not factored in evaluation
* Figure 5, end of the sentence, typo evaluators’ capabilities
* Please state the metrics used for calculating inter-rater reliability, e.g. Cohen Kappa etc. making sure it is consistent with other papers to be comparable

---

> ### Author Response · Authors · 2023-11-17
> **Response to Reviewer WsKe [1/2]**
>
> Thank you for the constructive feedback! We are encouraged that you recognize LLMBar as resolving a major missing piece in the high-quality, scalable meta-evaluation during the current surge of LLM evaluator usage. We appreciate your detailed comments and suggestions and will address them below in detail.
>
> **(1) Some confusion in how the evaluation set was generated.**
>
> Regarding “$O_1$ is generated by weaker models and $O_2$ by stronger models”, this is employed mainly for the collection of the Neighbor set. We had provided the details in footnote 5 (in the old version, now deleted) : 'We generate $O_1$ using an instruction-tuned LLaMA-7B, and $O_2$ is taken as the reference output from original datasets, which are generated by text-davinci-003 in Alpaca, by humans in OpenAssistant, and by ChatGPT in ShareGPT'. We appreciate your raised point and acknowledge that including this detail in a footnote may have caused confusion for readers. In the updated version, we have moved this detail into the main text of Section 2.2 for better clarity.
>
> **(2) Not sure what “manual filtering and modification” entail.**
>
> This term refers to the final stage in constructing both the Natural and Adversarial sets. In this process, each instance is carefully examined by the authors. If there is no objective preference between two outputs, or if the label is incorrect, we will then modify the instance accordingly or discard it if making such modifications is difficult. We have provided a more detailed explanation in the updated version of Section 2.1. For specific examples of this process, please check out Appendix A.1, A.2, and A.4.
>
> **(3) The human evaluators are co-authors of the papers, which can raise issues of objectivity compared with other crowd-sourced human evaluators.**
>
> We would like to clarify the procedure for conducting the human evaluation: For each instance, one author (annotator A) annotates the label, and another author (annotator B) annotates the instance again, without seeing the previous label. The annotator agreement rate is calculated as the percentage at which annotator A/B agree, which is also the human accuracy. To remain objective, the selected annotator B(s) have not seen the constructed data or annotator A’s labels before.
>
> We believe that this is a strength rather than a weakness. Our annotation task is objective by design and has very clear task definitions. The authors of the paper best understand the task instructions, and therefore, are best positioned to provide high-quality annotations. On the other hand, crowdworkers recruited through platforms like MTurk are more likely to introduce noise, such as misunderstanding task instructions. Due to this exact reason, several works in the past refer to author-annotated data as the “expert” test set, contrasting it with crowdsourced data that is assumed to be of lower quality but easier to collect at scale (Goyal et al., 2022).
>
> If the goal was to collect a diverse set of instructions or queries that reflect real-world usage distributions, then recruiting a diverse set of crowdworkers would be critical. However, in our setting, letting authors do the annotation and human evaluation is the best approach.

---

> ### Author Response · Authors · 2023-11-17
> **Response to Reviewer WsKe [2/2]**
>
> **(4) I am wondering whether it is a fair comparison to other benchmarks if the instructions are qualitatively different. For example, it is relatively easy to achieve high inter-rater reliability if the response is factual information rather than subjective.**
>
> We agree that LLMBar is qualitatively different from other benchmarks, and is designed to focus on one particular aspect of evaluation, i.e., instruction-following. We believe that it fills a critical gap in prior work. While studying previous meta-evaluation benchmarks (FairEval, LLMEval^2, MT-Bench), we observed that they cannot distinguish between different evaluators (see Figure 5) and fail to provide recommendations into which LLM evaluators are strongest. We hypothesize that one reason was that prior benchmarks cannot disentangle the effects of subjectivity and noise in their annotations, i.e., is the disagreement between annotators due to annotation errors/noise, or is it because the choice is inherently subjective? Therefore, they end up scoring most evaluators similarly.
>
> Our LLMBar benchmark addresses this gap in prior work by providing an **objective** (and also challenging) benchmark for comparing different evaluators. We intended to show such differences via our experiments and analysis. We show that
> + LLMBar is more objective and the annotation quality is higher, hence it achieves higher human agreement as shown in Section 4.2. The objectivity and high-quality annotation make our benchmark more reliable and indicative.
> + LLMBar is more challenging and distinguishes different evaluators better. This is demonstrated by Figure 5, where different evaluators perform similarly on other benchmarks but distinctly on LLMBar.
>
> We argue that our experiments deliver a fair comparison, as the goal is to show how challenging, objective, and high-quality our benchmark is compared to other benchmarks via the experiments. In the future, we hope that LLMBar can serve as a “stress test” to compare different evaluators, possibly in addition to other aspects (other than instruction following) that practitioners want to evaluate.
>
> **(5) It would be ideal to have another benchmark with comparable features (e.g., objectiveness and similar inter-rater agreement)**
>
> We believe that LLMBar is the **first** meta-evaluation benchmark that ensures objectivity and achieves an inter-annotator agreement of more than 90%. We focused on instruction following due to its importance in using LLMs. We hope that future work will build on the ideas in this work and introduce other objective benchmarks, possibly relating to other aspects apart from instruction following.
>
> **(6) Please state the metrics used for calculating inter-rater reliability, e.g. Cohen Kappa etc. making sure it is consistent with other papers to be comparable.**
>
> Our human agreement/accuracy is calculated as the percentage at which the two annotators agree with each other. In Section 4.2, we refer to the average human accuracy in FairEval and the agreement in MT-Bench is the average agreement among each pair of two annotators, so our calculation is mathematically equivalent to those of other papers we compared.
>
> **(7) Typo in the caption of Figure 5**
>
> Thanks for pointing it out! We have fixed it in the updated version.
>
> ## Reference
>
> Tanya Goyal, Junyi Jessy Li, and Greg Durrett. 2022. SNaC: Coherence Error Detection for Narrative Summarization. In *Proceedings of the 2022 Conference on Empirical Methods in Natural Language Processing*, pages 444–463, Abu Dhabi, United Arab Emirates. Association for Computational Linguistics.

---

> ### Author Response · Authors · 2023-11-21
> **Reminder of the Discussion Period Deadline**
>
> Dear Reviewer WsKe,
>
> We hope this message finds you well. As we're approaching the last day of the reviewer-author discussion period, we wanted to gently follow up regarding our previous response to your review. We value your input highly and would greatly appreciate it if you could let us know whether our rebuttal has addressed your initial concerns, which were primarily clarification questions. Your feedback is crucial for us, and we remain open to further discussion if needed.
>
> Thank you once again for your insightful review and the time you have invested in it.
>
> Best regards,
>
> Authors 7039

---

### Official Review · Reviewer_kxW7 · 2023-11-01

**Soundness:** 3 good
**Presentation:** 4 excellent
**Contribution:** 3 good
**Rating:** 8
**Confidence:** 3

**Summary:**

The paper proposes a challenging meta-evaluation benchmark, consisting of 100 natural and 319 adversarial <instruction, preferred output, non-preferred output> triplets, for evaluating instruction-following capabilities of LLMs. 100 natural instructions are carefully chosen from the existing benchmarks, AlpacaFarm and LLMEval. Adversarial set consists of 319 samples collected through 4 different strategies. 1) Uses neighbor search within the dataset to select a similar instruction, and then uses a strong LLM on the retrieved instruction to generate the non-preferred output. 2) Uses GPT-4 to generate a variant of instruction which is relevant but not similar and then uses the new instruction to generate the non-preferred output. 3) Uses GPT-4 to produce a superficially good but unhelpful response (non-preferred output). 4) Manual construction. The new dataset has very high inter-annotator agreement.

On the adversarial set they found that ChatGPT- and LLAMA-2-70B-based evaluators perform worse than the random chance. So, authors also propose new prompting strategies: Rules (a list of general rules to follow), Metrics (prompt LLM to generate a set of instruction-specific metrics and use them to evaluate the outputs) and Swap (generate scores for both the outputs orders o1,o2 and o2,o1; use LLM to generate final score by using combinations of both responses if they are contradictory). Their new prompting strategies improve the LLM-based evaluator's performance.

**Strengths:**

- A possibly useful benchmark for evaluating instruction-following capabilities of LLM.

- Useful insights on the capabilities of LLM-based evaluators (e.g., Llama-based and Chat-GPT-based evaluators incompetency in evaluating instruction-following capabilities of LLMs.)

- Proposed prompting strategies improve evaluator's performance.

**Weaknesses:**

Are LLMs likely to sample non-preferred output for instructions in the adversarial pool? Could you provide the distribution of generating preferred and non-preferred outputs for a few LLMs? It is unclear, whether the findings of the paper (or to say, improving performance of evaluator-LLMs on adversarial samples) would actually lead to improved reliability of LLM-based evaluations.

**Questions:**

NA

---

> ### Author Response · Authors · 2023-11-17
> **Response to Reviewer kxW7**
>
> Thank you for the constructive feedback! As LLM evaluators become more commonly adopted, we are encouraged by your appreciation of the contributions that LLMBar and our prompting strategies provide to the research community. We believe LLMBar will contribute to standardized meta-evaluation for future LLM evaluators.
>
> To address your concern **Are LLMs likely to sample non-preferred output for instructions in the adversarial pool? Could you provide the distribution of generating preferred and non-preferred outputs for a few LLMs?**:
> + We randomly sampled 50 instructions from Adversarial and found that ChatGPT (`gpt-3.5-turbo-0613`), LLaMA-2-70B-Chat, and PaLM2 (`text-bison-001`) can faithfully follow 90%, 86%, and 84% of them respectively by human evaluation.
> + Even though the LLM can generate instruction-following outputs, it does not mean the LLM can discern instruction-following outputs. This is shown in our results and echoed by West et al., 2023, which shows that being able to generate does not entail being able to understand. In this work, we focus on studying whether the LLM can distinguish instruction-following outputs from dispreferred outputs. The adversarial instances are constructed such that it is challenging for LLM evaluators to identify the instruction-following outputs, rather than being deliberately hard to generate based on the instruction. We have added this clarification in Section 2.2 of our updated version. Due to the page limit, we plan to discuss this point to provide more insights in our next version.
> + Even GPT-4 is likely to generate outputs not following simple instructions (Wu et al., 2023; Li et al., 2023). This is a key motivation for our focus on evaluating whether LLM evaluators can detect instruction-following outputs, an effort that can facilitate future development of instruction-following models. For example, in the Neighbor set, there is an instance asking for **methods of learning** search engine optimization (SEO). The dispreferred output deviates by providing **specific ways of doing SEO**. We observed that GPT-4, when prompted with this instruction, tends to produce outputs similar to the dispreferred one.
>
> In response to **whether the findings will improve LLM evaluators' ability**: Though LLMBar’s Adversarial set has a distinct distribution, we believe that an improvement on it indicates a more reliable LLM evaluator in the real-world distribution – being able to distinguish instruction following from other superficial appeal should be a basic requirement for LLM evaluators, and failing to do so makes the evaluator misleading. However, we acknowledge the distribution shift and that the Adversarial set is more like a “stress test”, hence we also provide the Natural set as a reference. We see that the performance trends on the Natural set generally agree with that on the Adversarial set but with a smaller margin.
>
> ## Reference
>
> Wu, Zhaofeng, et al. "Reasoning or reciting? exploring the capabilities and limitations of language models through counterfactual tasks." *arXiv preprint arXiv:2307.02477* (2023).
>
> Li, Shiyang, et al. "Instruction-following evaluation through verbalizer manipulation." *arXiv preprint arXiv:2307.10558* (2023).
>
> West, Peter, et al. "The Generative AI Paradox:" What It Can Create, It May Not Understand"." *arXiv preprint arXiv:2311.00059* (2023).

---

### Author Response · Authors · 2023-11-17
**General Response**

We sincerely thank all the reviewers for their constructive comments and suggestions. We are also delighted to see that the reviewers generally have a very high appreciation for our work's novelty, experiments, potential impact, and presentation.

We have updated a new version of our paper and added the following experiments:
+ We evaluated two reward models and one preference model on LLMBar. The results are presented in the newly added Section 4.5. We found that these three models fall significantly short on LLMBar, even on the Natural set, suggesting that current reward models and preference models struggle to identify instruction-following outputs.
+ We evaluated PaLM2-based evaluators and added the results. We found that PaLM2 is a good base LLM for evaluation, generally outperforming all the other evaluated LLMs except for GPT-4.

We hope the added experiments will provide more insights.

Meanwhile, we have also gained constructive feedback from all the concerns and questions raised. To address them, we have updated our paper in the following ways (and we highlighted relevant revisions in red color):
+ We fixed all the typos the reviewers pointed out.
+ We improved our presentation (including text, tables, and figures) and added some detailed explanations for better clarity.

For other concerns, we addressed them in our responses.

---

### Meta-Review · Area_Chair_S8fV · 2023-12-11

**Metareview:**

This paper tries to address one important problem on how to assess the of the LLM, particularly on the instruction following. It provides a carefully curated dataset that is potentially useful for "stress-testing" the LLM evaluators.

Strength:
1. A potentially useful dataset for evaluating the LLM evaluators.
2. Provides a set of new prompting strategies.

Weakness:
1. Human evaluators are co-authors, which could provide some hidden bias that is difficult to detect.
2. The evaluated results of capabilities using LLMBar over existing LLMs (which is good or bad) seem to be not surprising. How this would provide further insights on improving the LLM models is not very clear.

**Justification For Why Not Higher Score:**

See weakness.

**Justification For Why Not Lower Score:**

A potentially useful dataset for evaluating the instruction following capacities of the LLMs.

---

### Decision · Program_Chairs · 2024-01-16

Accept (poster)